# Towards False-claim-resistant Model Ownership Verification via Targeted Fingerprint

## Abstract

The utilization of open-source pre-trained models has become a prevalent practice, but unauthorized reuse of pre-trained models may pose a threat to the intellectual property rights (IPR) of the model developers. Model fingerprinting, which does not necessitate modifying the model to verify whether a suspicious model is reused from the source model, stands as a promising approach to safeguarding the IPR. In this paper, we revisit existing model fingerprinting methods and demonstrate that they are vulnerable to false claim attacks where adversaries falsely assert ownership of any third-party model. We reveal that this vulnerability mostly stems from their *untargeted* nature, where they generally compare the outputs of given samples on different models instead of the similarities to specific references. Motivated by these findings, we propose a targeted fingerprinting paradigm (*i.e.*, FIT-Print) to counteract false claim attacks. Specifically, FIT-Print transforms the fingerprint into a targeted signature via optimization. Building on the principles of FIT-Print, we develop bit-wise and list-wise black-box model fingerprinting methods, *i.e.*, FIT-ModelDiff and FIT-LIME, which exploit the distance between model outputs and the feature attribution of specific samples as the fingerprint, respectively. Extensive experiments on benchmark models and datasets verify the effectiveness, conferrability, and resistance to false claim attacks of our FIT-Print.

## 1 Introduction

Deep learning models, especially deep neural networks (DNNs), have been widely and successfully deployed in widespread applications (Wang et al., 2024; Li et al., 2022a; He et al., 2023). In general, obtaining a well-performed model requires considerable computational resources and human expertise and is, therefore, highly expensive. In particular, some models are released to the open-source community (*e.g.*, Hugging Face) for academic or educational purposes. However, the development of model reuse techniques, such as fine tuning (Liu et al., 2018) and transfer learning (Zhuang et al., 2020), poses a potential threat to the intellectual property rights (IPR) of these models. With these methods, malicious developers can easily reuse open-source models for commercial purposes without authorization. How to protect their IPR becomes a vital problem.

Currently, ownership verification stands as a widely adopted post-hoc approach for safeguarding the IPR of model developers. This method intends to justify whether a suspicious third-party model has been reused from the protected model (Zhang et al., 2020b; Li et al., 2022d; Sun et al., 2023). Existing techniques to implement ownership verification can be broadly categorized into two main types: model watermarking and model fingerprinting. Model watermarking (Adi et al., 2018; Li et al., 2022d; Yang et al., 2023) involves embedding an owner-specific signature (*i.e.*, watermark) into the model. The model developer can extract the watermark inside the model to verify its ownership. On the contrary, model fingerprinting (Cao et al., 2021; Li et al., 2021; Jia et al., 2022) aims to identify the intrinsic feature (*i.e.*, fingerprint) of the model instead of modifying the protected model. The fingerprint can be represented as the outputs of some testing samples at a particular mapping function. Comparing model fingerprints enables comparing the source and suspicious models to examine whether the latter is a reused version of the former. Arguably, model fingerprinting is more convenient and feasible than model watermarking since the former does not necessitate any alteration to the parameters, structure, and training procedure and thus has no negative impact on the model.

In this paper, we reveal that existing model fingerprinting methods, no matter whether they are bit-wise (Li et al., 2021) or list-wise (Jia et al., 2022) (*i.e.*, extract the fingerprint bit by bit or as a

Figure 1: The comparison of untargeted and targeted fingerprinting paradigms. Untargeted methods generally compare the output of given samples. Accordingly, using some transferable samples can lead to false claims. Targeted fingerprinting calculates the similarity to a specific signature, which restricts the fingerprint space around the target and, therefore, mitigates false claim attacks.

whole list), are vulnerable to false claim attacks[1]. In general, false claim attacks allow adversaries to falsely assert to have the ownership of third-party models that is *not* a reused model by creating a counterfeit ownership certificate (*i.e.*, watermark or fingerprint). In particular, false claim attacks can be treated as finding transferable ownership certificates across models since registering the certificate with a timestamp can avoid any false claim with a later timestamp.(Liu et al., 2024). We show that the adversary can conduct false claim attacks by constructing transferably 'easy' samples that can be correctly classified with high confidence (as shown in Section 2.3). Existing fingerprinting methods tend to compare the outputs of testing samples, and these elaborated easy samples can have similar high-confident outputs on various models, thus leading to independent models being misjudged as reused models. We argue that this vulnerability mostly stems from the *untargeted* nature of existing methods. Specifically, they generally compare the outputs of any given samples on different models instead of the similarities to specific references. The untargeted nature enlarges the space of viable fingerprints. It makes adversaries easily find alternative transferable samples that have similar output on independent models, as illustrated in Figure 1.

Motivated by the aforementioned understandings, we introduce a new fingerprinting paradigm, dubbed False-claIm-resistant Targeted model fingerPrinting (FIT-Print), where the fingerprint comparison is *targeted* instead of untargeted. Specifically, we optimize the perturbations on the testing samples to make the output of the fingerprinting mapping function close to a specific signature (*i.e.*, the target fingerprint). It restricts the (potential) fingerprint space and significantly reduces the probability of a successful false claim attack. Based on our FIT-Print, we design two targeted model fingerprinting methods, including FIT-ModelDiff and FIT-LIME, as the representatives of bit-wise and list-wise methods, respectively. FIT-ModelDiff exploits the distances between outputs, while FIT-LIME leverages the feature attribution of testing samples as the fingerprint.

Our main contributions are four-fold: **(1)** We revisit existing model fingerprinting methods and reveal that existing methods are vulnerable to false claim attacks. **(2)** We introduce a new fingerprinting paradigm (*i.e.*, FIT-Print), where we conduct verification in a targeted manner with a given reference. **(3)** Based on our FIT-Print, we design two black-box targeted model fingerprinting methods: FIT-ModelDiff and FIT-LIME. **(4)** We conduct experiments on benchmark datasets to verify the effectiveness and conferrability of FIT-Print, and its resistance to false claims and adaptive attacks.

## 2 REVISITING EXISTING MODEL FINGERPRINTING METHODS

### 2.1 THREAT MODEL OF MODEL FINGERPRINTING

Following prior works (Liu et al., 2024; Waheed et al., 2024), there are three parties in the threat model, including the *model developer*, the *model reuser*, and the *verifier*.

**Assumptions of the Model Developer and Verifier.** The model developer is the true owner of the source model and can register its model and fingerprint to the trustworthy verifier with a timestamp. The verifier is responsible for fingerprint registration and verification. In case the model is reused by a model reuser, the model developer can ask the verifier for ownership verification. In particular, if two

---

[1]The concept and definition of false claim attacks was initially introduced in (Liu et al., 2024) and primarily targeted at attacking model watermarking methods.

parties can simultaneously provide fingerprints and verify the ownership of a model, the fingerprint with a later timestamp will be deemed invalid. The model developer and the verifier are assumed to have **(1)** white-box access to its source model and **(2)** black-box access to the suspicious model.

**Assumptions of the Model Reuser.** Model reusers aim to avoid having their authorized reuse detected by the verifier. To achieve this, they can first modify the victim model via various techniques, such as fine-tuning, pruning, transfer learning, and model extraction, before deployment.

## 2.2 THE FORMULATION OF EXISTING FINGERPRINTING METHODS

In this section, we primarily focus on black-box instead of white-box model fingerprinting methods since they are more practical in real-world applications. We outline the formulations of these methods to aid in the analysis and design process in the subsequent sections of this paper and follow-up research. In general, existing black-box model fingerprinting methods can be categorized into two types: adversarial example-based (AE-based) fingerprinting methods and testing-based fingerprinting methods. We also include a broader discussion about other model fingerprinting in Appendix L.

**AE-based Fingerprinting Methods.** AE-based fingerprinting methods (Cao et al., 2021; Lukas et al., 2021; Pautov et al., 2024) assume that the independent model has a unique decision boundary. Based on this assumption, they exploit adversarial examples (AE) (Ren et al., 2020; Wan et al., 2024) to characterize the properties of the decision boundary of a model. AE-based fingerprinting methods validate whether the AEs are misclassified by the source model and the suspicious model. If so, the suspicious model can be treated as a reused version of the source model. The proposition tested in AE-based methods can be formulated as follows.

**Proposition 1** (Ownership Verification of AE-based Methods). *Let $M_o$ be the source model and $M_s$ be the suspicious model, and $g(\boldsymbol{x})$ is the function that always outputs the ground-truth label of any input data $\boldsymbol{x}$. If for any testing sample $\boldsymbol{x} \in \mathcal{X}_T$ ($\mathcal{X}_T$ denotes the set of the testing samples) we have*

$$M_o(\boldsymbol{x}) = M_s(\boldsymbol{x}) \neq g(\boldsymbol{x}), \tag{1}$$

*the suspicious model $M_s$ can be asserted as a reused version of the source model $M_o$.*

**Testing-based Fingerprinting Methods.** Testing-based fingerprinting methods (Li et al., 2021; Jia et al., 2022; Guan et al., 2022) aim to compare the suspicious model with the source model on a specific mapping function $f(\cdot)$. If the outputs of these two models are similar, the suspicious model can be regarded as being reused from the source model. As such, the core of testing-based fingerprinting methods is how to design the mapping function $f(\cdot)$. The proposition used in testing-based methods can be formulated as follows.

**Proposition 2** (Ownership Verification of Testing-based Methods). *Let $M_o$ be the source model and $M_s$ be the suspicious model. If for a specific mapping function $f(\cdot)$ and any testing sample $\boldsymbol{x} \in \mathcal{X}_T$ ($\mathcal{X}_T$ is the set of the testing samples) we have*

$$\frac{1}{|\mathcal{X}_T|} \sum_{\boldsymbol{x} \in \mathcal{X}_T} \mathtt{dist}(f[M_o(\boldsymbol{x})], f[M_s(\boldsymbol{x})]) \leq \tau, \tag{2}$$

*where $\tau$ is a small positive threshold and $\mathtt{dist}(\cdot, \cdot)$ is a distance function, the suspicious model $M_s$ can be asserted as reused from the source model $M_o$.*

## 2.3 THE FALSE CLAIM ATTACK AGAINST MODEL FINGERPRINTING

Existing model fingerprinting methods primarily assume that the model reuser is the adversary while paying little attention to the false claim attack (Liu et al., 2024) where the model developer is the adversary. The definition of the false claim attack is as follows.

**Definition 1.** *A false claim attack refers to a malicious attempt by a malicious model developer to falsely assert the ownership of an independent model $M_I$ by registering some fraudulent testing samples $\bar{\boldsymbol{x}}$ that can pass the ownership verification of Proposition 1 or Proposition 2.*

Some terms (*e.g.*, ambiguity attack and false positive rate) may have a similar definition to the false claim attack. We clarify their differences in Appendix M. Since registering the fingerprint with a timestamp can prevent any false claim after registration, the success of false claim attacks

Table 1: False claim attack against three testing-based methods. It is observed that the distances between independent models and the source model (Ind. Model Dist.) after the attack are approximately equal to or less than the average distance between the reused models and the source model (Reused Model Dist.), demonstrating the vulnerability of existing methods against false claim attacks.

| | Method→ | ModelDiff | | Zest | | SAC | |
|---|---|---|---|---|---|---|---|
| | Dataset→ | SDogs120 | Flowers102 | SDogs120 | Flowers102 | SDogs120 | Flowers102 |
| Ind. Model Dist. | Before Attack | 0.131 | 0.114 | 0.177 | 0.161 | 0.080 | 0.094 |
| | After Attack | 0.093 | 0.083 | 0.114 | 0.098 | 0.078 | 0.092 |
| Reused Model Dist. | Average | 0.108 | 0.092 | 0.095 | 0.072 | 0.079 | 0.081 |

hinges on generating a transferable fingerprint. For AE-based methods, Liu et al. (2024) have successfully implemented the false claim attacks by constructing transferable AEs. As such, we focus on implementing false claim attacks against the cutting-edge testing-based methods. Our primary insight is to craft inverse-AEs $\bar{x}$ which can be 'easily' classified, leading to

$$M_o(\bar{x}) \approx M_I(\bar{x}) \Rightarrow \texttt{dist}(f[M_o(\bar{x})], f[M_I(\bar{x})]) \approx 0 \leq \tau. \tag{3}$$

To execute this strategy, motivated by fast gradient sign method (FGSM) (Goodfellow et al., 2015) for AE generation, we propose to leverage Eq. (4) to generate malicious fingerprinting samples.

$$\bar{x} = x - \gamma \cdot \texttt{sign}(\nabla J(M_o, x, y)), \tag{4}$$

where $\texttt{sign}(\cdot)$ denotes the sign function, $J(\cdot)$ represents the loss function associated with the original task of $M_o$, and $\gamma$ signifies the magnitude of the perturbation. More powerful transferable adversarial attacks can be exploited here but we aim to show that using simple FGSM can also falsely claim to have ownership of some independent models. We exploit three representative testing-based methods, ModelDiff (Li et al., 2021), Zest (Jia et al., 2022), and SAC (Guan et al., 2022), to validate the effectiveness of our false claim attacks. The complete results can be found in Appendix E. As shown in Table 1, SAC is poor at identifying models of the same tasks, even without attacks. Moreover, after attacks, the distances between the source model $M_o$ and the independent model $M_I$ of all three methods are approximately equal to or less than the average distances between reused models and the source model. It indicates that $M_I$ will be asserted as reused from $M_o$, which is a false alarm. The results demonstrate that existing model fingerprinting methods are vulnerable to false claim attacks.

## 3 THE PROPOSED METHOD

### 3.1 DESIGN OBJECTIVES

Following prior works (Lukas et al., 2021; Liu et al., 2024), the objectives of model fingerprinting methods can be summarized as effectiveness, conferrability, and resistance to false claim attacks.

- **Effectiveness:** Effectiveness means that the model developer can successfully verify the ownership of the source model through the model fingerprinting method.

- **Conferrability:** Conferrability is defined to ensure that the model fingerprint needs to be conferable to the models that are reused from the source model. In other words, the fingerprints of the reused models and the source model need to be similar.

- **Resistance to False Claim Attacks:** It requires that the fingerprints of independently trained models need to be different. Also, a malicious model developer cannot construct a transferable fingerprint that can be extracted from independently trained models.

### 3.2 THE INSIGHT OF OUR FIT-PRINT

As discussed in Section 2.3, existing model fingerprinting methods are vulnerable to false claim attacks. We argue that the vulnerability stems primarily from the 'untargeted' characteristic of the fingerprinting methods. The untargeted characteristic leads to a large fingerprint space that can accommodate transferable adversarial fingerprints. In this paper, we propose FIT-Print, a targeted model fingerprinting framework to mitigate false claim attacks. Our main insight is that although it is tough to find the space that can only transfer among reused models, we can turn the fingerprint into a target one to restrict the fingerprinting space and reduce the adversarial transferability of fingerprints.

Figure 2: The pipeline of FIT-Print. In testing sample extraction, FIT-Print optimizes the perturbations to turn the fingerprint vector close to the target fingerprint. In the ownership verification stage, FIT-Print extracts the fingerprint from the suspicious model and compares it with the original fingerprint.

Given a mapping function $f(\cdot)$ and a target fingerprint $\boldsymbol{F}$, our goal is to make the fingerprint vector $\boldsymbol{v} = f(M_s(\boldsymbol{x}))$ to be close to $\boldsymbol{F}$. Thus, the proposition of FIT-Print can be defined as follows.

**Proposition 3.** *Let $M_s$ be the suspicious model. If for a specific mapping function $f(\cdot)$ and testing sample $\boldsymbol{x} \in \mathcal{X}_T$ ($\mathcal{X}_T$ is the set of the testing samples), we have*

$$\frac{1}{|\mathcal{X}_T|} \sum_{\boldsymbol{x} \in \mathcal{X}_T} \texttt{dist}(f[M_s(\boldsymbol{x})], \boldsymbol{F}) \leq \tau, \tag{5}$$

*where $\tau$ is a small threshold and $\texttt{dist}(\cdot, \cdot)$ is a distance function, the suspicious model $M_s$ can be asserted as reused from the owner of the fingerprint $\boldsymbol{F}$.*

In FIT-Print, we assume that the target fingerprint $\boldsymbol{F} \in \{-1, 1\}^k$ is a binary vector consisting of $-1$ or $1$, and we can get the output logits of $M_s(\boldsymbol{x})$. The discussion on the label-only scenario where we can only get the Top-1 label can be found in Appendix G. We assume that the target fingerprint cannot be arbitrarily chosen and needs to be registered with a third-party institution. As shown in Fig. 2, FIT-Print can be divided into two stages: testing sample extraction and ownership verification. The technical details are described as follows.

### 3.3 TESTING SAMPLE EXTRACTION

In the testing sample extraction stage, we aim to find the optimal testing sample set $\mathcal{X}_T$ to make any reused models satisfy Eq. (5) in Proposition 3. Therefore, in FIT-Print, we first initialize the testing samples $\mathcal{X}_T$ and the corresponding perturbations $\mathcal{R}$. We denote the $i$-th element in $\mathcal{X}_T$ and $\mathcal{R}$ as $\boldsymbol{x}_i$ and $\boldsymbol{r}_i$ respectively. The element $\boldsymbol{x}_i$ is set to an initial value $\boldsymbol{x}_i^0$ and we can initialize the testing samples to any images. The testing samples in $\mathcal{X}_T$ can be constructed by adding the perturbations to the initial values, *i.e.*, $\boldsymbol{x}_i = \boldsymbol{x}_i^0 + \boldsymbol{r}_i$. After that, we need to optimize the perturbations $\mathcal{R}$ to make the fingerprint vector $\boldsymbol{v}$ close to the target fingerprint $\boldsymbol{F}$. We can define the testing sample extraction as an optimization problem, which can be formalized as follows.

$$\min_{\mathcal{R} = \{\boldsymbol{r}_1, \dots, \boldsymbol{r}_{|\mathcal{R}|}\}} \frac{1}{|\mathcal{X}_T|} \sum_{i=1}^{|\mathcal{X}_T|} [\mathcal{L}(f(M_o(\boldsymbol{x}_i^0 + \boldsymbol{r}_i), \boldsymbol{F}) + \lambda \cdot \|\boldsymbol{r}_i\|_2], \tag{6}$$

where $\|\cdot\|_2$ calculates the $\ell_2$-norm. The first term in Eq. (6) quantifies the dissimilarity between the output fingerprint vector $\boldsymbol{v}$ and the target fingerprint $\boldsymbol{F}$. The second term regularizes the extent of the perturbations $\mathcal{R}$. We utilize the hinge-like loss (Fan et al., 2019) as $\mathcal{L}(\cdot)$, as follows.

$$\mathcal{L}(\boldsymbol{v}, \boldsymbol{F}) = \sum_{i=1}^{k} \max(0, \varepsilon - \boldsymbol{v}_i \cdot \boldsymbol{F}_i). \tag{7}$$

In Eq. (7), $\boldsymbol{v}$ is the fingerprint vector, where $\boldsymbol{v}_i = f[M_s(\boldsymbol{x}_i)]$, and $\varepsilon$ is the control parameter. $\boldsymbol{F}_i$ is the $i$-th element in $\boldsymbol{F}$. Optimizing Eq. (7) can make the signs of the corresponding elements in $\boldsymbol{v}$ and $\boldsymbol{F}$ the same. Moreover, inspired by the insight of (Lukas et al., 2021), we craft some augmented models by applying model reuse techniques (*e.g.*, fine-tuning, pruning, or transfer learning) and exploit them to extract the fingerprint to improve the conferrability of FIT-Print. The set of augmented models is denoted as $\mathcal{M}$. The loss function with augmented models can be defined as Eq. (8).

$$\min_{\mathcal{R} = \{\boldsymbol{r}_1, \dots, \boldsymbol{r}_{|\mathcal{R}|}\}} \frac{1}{|\mathcal{M}| \cdot |\mathcal{X}_T|} \sum_{M \in \mathcal{M}} \sum_{i=1}^{|\mathcal{X}_T|} [\mathcal{L}(f(M(\boldsymbol{x}_i^0 + \boldsymbol{r}_i), \boldsymbol{F}) + \lambda \cdot \|\boldsymbol{r}_i\|_2]. \tag{8}$$

By optimizing Eq. (8), we can get the optimal testing samples that are conferrable to reused models and the model developer can afterward utilize them to verify the ownership.

### 3.4 OWNERSHIP VERIFICATION

In the ownership verification stage, given a suspicious model $M_s$, FIT-Print examines whether the suspicious model $M_s$ is reused from the source model $M_o$ by justifying whether $M_s$ satisfies Eq. (5). Specifically, we first calculate the fingerprint vector $\tilde{v}$ of the suspicious model $M_s$ using the extracted testing samples in $\mathcal{X}_T$. Each element $\tilde{v}_i = f(M_s(x_i^0 + r_i))$. Since optimizing Eq. (8) makes the signs of the fingerprint vector $\tilde{v}$ represent the fingerprint $\tilde{F}$ of the model, we need to transform $\tilde{v}$ into a binary vector by applying the sign function $\texttt{sign}(\cdot)$ to get $\tilde{F}$, as Eq. (9).

$$\tilde{F}_i = \texttt{sign}(\tilde{v}_i) = \begin{cases} 1, & \tilde{v}_i \geq 0 \\ -1, & \tilde{v}_i < 0 \end{cases}. \tag{9}$$

Subsequently, we leverage the bit error rate (BER) as the distance function $\texttt{dist}(\cdot)$ in Eq. (5) and the BER is the distance between the extracted fingerprint and the target fingerprint, as follows.

$$\texttt{BER} = \frac{1}{k}\sum_{i=1}^{k} \mathbb{I}\{\tilde{F}_i \neq F_i\}, \tag{10}$$

where $k$ is the length of the fingerprint and $\mathbb{I}\{\cdot\}$ is the indicator function. As Proposition 3, if the BER is lower than the threshold $\tau$, the suspicious model $M_s$ can be asserted as a reused model. For choosing the threshold $\tau$ to resist false claim attacks, we have the following Theorem 1.

**Theorem 1.** *Given the security parameter $\kappa$ and the fingerprint $F \in \{-1, 1\}^k$, if $\tau$ satisfy that*

$$\sum_{d=0}^{\lfloor \tau k \rfloor} \binom{k}{d}(\frac{1}{2})^k \leq \kappa, \tag{11}$$

*where $\binom{k}{d} = k!/[d!(k-d)!]$, the probability of a false alarm, i.e., the BER is less than $\tau$ with random testing samples, is less than $\kappa$.*

The proof of Theorem 1 can be found in Appendix B. We also conduct an empirical evaluation on the resistance of FIT-Print against adaptive false claim attacks in Section 4.4.

### 3.5 DESIGNING THE MAPPING FUNCTION IN FIT-PRINT

Section 3.3-3.4 introduce the stages of FIT-Print. Arguably, its key is how to design the mapping function $f(\cdot)$. Inspired by Li et al. (2021) and Shao et al. (2024a), we leverage the paradigm of FIT-Print and design two targeted model fingerprinting methods, including FIT-ModelDiff and FIT-LIME, as the representatives of bit-wise and list-wise methods, respectively.

#### 3.5.1 FIT-MODELDIFF

FIT-ModelDiff is a bit-wise fingerprinting method that extracts the fingerprint bit by bit. The main insight of FIT-ModelDiff is to compare the distance between the output logits of perturbed samples $x_i^0 + r_i$ and benign samples $x_i^0$. The vector of the distances is called the decision distance vector (DDV). Given the suspicious model $M_s$, DDV can be calculated as follows:

$$\texttt{DDV}_i = \texttt{cos\_sim}(M_s(x_i^0 + r_i), M_s(x_i^0)) = \frac{M_s(x_i^0 + r_i) \cdot M_s(x_i^0)}{\|M_s(x_i^0 + r_i)\| \cdot \|M_s(x_i^0)\|}, \tag{12}$$

where $\texttt{DDV}_i$ represents the $i$-th element in the DDV and $\texttt{cos\_sim}(\cdot, \cdot)$ is the cosine similarity function. Since the output logits after softmax of the model $M_s$ is always positive, the range of the DDV is $[0, 1]$. As proposed in Section 3.3, we aim to make the sign of the fingerprint vector $v$ to be the same as the target fingerprint $F$. Therefore, to achieve this goal, we need to subtract a factor from DDV to make the range of $v$ including both positive and negative values, as Eq. (13).

$$v_i = f(M_s(x_i^0 + r_i), M_s(x_i^0)) = \texttt{DDV}_i - \cos(\alpha) = \frac{M_s(x_i^0 + r_i) \cdot M_s(x_i^0)}{\|M_s(x_i^0 + r_i)\| \cdot \|M_s(x_i^0)\|} - \cos(\alpha), \tag{13}$$

where $\cos(\cdot)$ is the cosine function and $\alpha$ is the bias parameter. The final fingerprint vector $v$ can be used for testing sample extraction or ownership verification.

### 3.5.2 FIT-LIME

FIT-LIME is a list-wise method that extracts the fingerprint as a whole list. FIT-LIME implements the mapping function $f(\cdot)$ via a popular feature attribution algorithm, local interpretable model-agnostic explanation (LIME) (Ribeiro et al., 2016). LIME outputs a real-value importance score for each feature in the input sample $x$. We enhance the LIME algorithm and develop FIT-LIME to better cater to the needs of ownership verification. The details of FIT-LIME are elaborated as follows.

The first step of FIT-LIME is to generate $c$ samples that are neighboring to the input image $x$. We also gather the adjacent pixels in the image into a superpixel. Instead of using a clustering algorithm, we uniformly segment the input space into $k$ superpixels, where $k$ is the length of the targeted fingerprint. Assuming that $k = \mu \times \nu$, the image can be divided into $\mu$ rows and $\nu$ columns. Then, we randomly generate $c$ masks where each mask is a $k$-dimension binary vector, constituting $A \in \{0,1\}^{c \times k}$. Each element in each row of the matrix $A$ corresponds to a superpixel in the image $x$. After that, we exploit the binary matrix to mask the image $x$ and generate the masked examples $\mathcal{X}^m$. If the element in the $i$-th row of the mask $A$ is 1, the corresponding superpixel preserves its original value. Otherwise, the superpixel is aligned with 0. Each row of the mask can generate a masked sample and the $c$ masked samples constitute the masked sample set $\mathcal{X}^m$.

The second step is to evaluate the output of the masked samples $\mathcal{X}^m$ on the suspicious model $M_s$. Different from primitive LIME, we utilize the entropy of the outputs so that it no longer depends on the label of $x$. The intuition is that if the important features are masked, the prediction entropy will significantly increase. Following this insight, we calculate the following equation in this step.

$$p_i = H[M_s(\mathcal{X}_i^m)], \tag{14}$$

where $H(\cdot)$ calculates the entropy, $p_i$ is the $i$-th element in $p$, and $\mathcal{X}_i^m$) is the $i$-th masked samples.

After that, the final step is to fit a linear model and calculate the importance score of each superpixel. The importance scores can be calculated via Eq. (15). The importance score vector will be used as the fingerprint vector $v$ in testing sample extraction and ownership verification.

$$v = (A^T A)^{-1} A^T p. \tag{15}$$

## 4 EXPERIMENTS

In this section, we evaluate the effectiveness, conferrability, and resistance to the false claim attack of our FIT-Print methods. We also include ablation studies on the hyper-parameters in FIT-Print. More experiments about the resistance to adaptive attacks, FIT-Print with different targets, initializations, and different numbers of augmented models are shown in Appendix F. We also discuss applying FIT-Print in the label-only scenario and to other models and datasets in Appendix G and J. The analysis of the overhead of FIT-ModelDiff and FIT-LIME can be found in Appendix H.

### 4.1 EXPERIMENTAL SETTINGS

**Models and Datasets.** Following prior works (Li et al., 2021; Jia et al., 2022), we utilize two widely-used convolutional neural network (CNN) architectures, MobileNetV2 (Sandler et al., 2018) (mbnetv2 for short) and ResNet18 (He et al., 2016), in our experiments. We train MobileNetv2 and ResNet18 using two different datasets, Oxford Flowers 102 (Flowers102) (Nilsback & Zisserman, 2008) and Stanford Dogs 120 (SDogs120) (Khosla et al., 2011), in total 4 source models. We primarily focus on image classification models in our experiments. In particular, we provide a case study about implementing FIT-Print to text generation models in Appendix J.3.

**Model Reuse Techniques.** We evaluate FIT-Print against the following five categories of model reuse techniques, including copying, fine-tuning, pruning, model extraction, and transfer learning. We further consider different implementations of these model reuse techniques in various settings and scenarios. For each source model, we train and craft three fine-tuning models, three pruning models, two extraction models, and three transfer learning models. These 12 models constitute the set of reused models. When experimenting on one source model, the other 36 models that are reused from other source models are treated as independent models. More details can be found in Appendix C.

**Baseline Methods.** We consider both AE-based and testing-based methods. For the former, we implement two typical methods, IPGuard (Cao et al., 2021) and MetaV (Pan et al., 2022). While

Table 2: Successful ownership verification rates of different model fingerprinting methods. '#Models' denotes the number of reused models and the 'N/A' indicates that the method can not be applied to detect this type of model reuse technique. Our FIT-ModelDiff and FIT-LIME outperform existing black-box methods and their performance is even on par with that of white-box ModelGiF. In particular, we mark failed cases (*i.e.*, $< 80\%$ or 'N/A') in red. Moreover, the BERs of FIT-Print are all 0.0%, indicating that FIT-Print does not lead to false alarms.

| Reuse Task↓ | #Models↓ | AE-based | | Testing-based | | | White-box | FIT-Print | |
|---|---|---|---|---|---|---|---|---|---|
| | | IPGuard | MetaV | ModelDiff | Zest | SAC | ModelGiF | FIT-ModelDiff | FIT-LIME |
| Copying | 4 | 100% | 100% | 100% | 100% | 100% | 100% | 100% | 100% |
| Fine-tuning | 12 | 100% | 100% | 100% | 100% | 100% | 100% | 100% | 100% |
| Pruning | 12 | 100% | 100% | 100% | 91.67% | 100% | 100% | 100% | 100% |
| Extraction | 8 | 50% | 87.5% | 50% | 25% | 100% | 100% | 100% | 100% |
| Transfer | 12 | N/A | N/A | 100% | N/A | 0% | 100% | 100% | 100% |
| Independent | 144 | 30.6% | 4.8% | 4.0% | 7.6% | 39.6% | 0.0% | 0.0% | 0.0% |

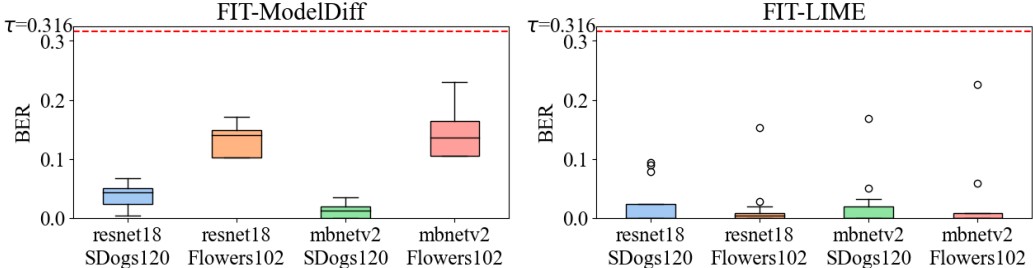

Figure 3: The BERs of different source models and their reused models with FIT-ModelDiff and FIT-LIME. The BERs are all less than the threshold $\tau$ marked with a red dashed line, indicating that FIT-ModelDiff and FIT-LIME can successfully recognize the reused models.

for the latter, we take three different methods, ModelDiff (Li et al., 2021), Zest (Jia et al., 2022), and SAC (Guan et al., 2022) as the baseline methods. We also include a state-of-the-art (SOTA) white-box model fingerprinting method, *i.e.*, ModelGiF (Song et al., 2023), for reference.

**Target Fingerprint.** As default, we select a logo of a file and a pen as the targeted fingerprint $\boldsymbol{F}$. We set the default length $k$ of the fingerprint $\boldsymbol{F}$ to be 256 and thus $\boldsymbol{F}$ is resized to $16\times16$. We set the security parameter $\kappa = 10^{-9}$. According to Eq. (11), the threshold $\tau$ is 0.316 in our experiments.

## 4.2 Evaluation on Effectiveness and Conferrability

Table 2 illustrates the percentage of successfully identified reused models (ownership verification rate). Both FIT-ModelDiff and FIT-LIME can recognize the reused models under five reuse techniques with 100% ownership verification rates, which outperform existing fingerprinting methods and perform on par with the SOTA white-box method, ModelGiF. Also, FIT-ModelDiff and FIT-LIME achieve 0.0% ownership verification rates on the independent models, indicating that our methods do not lead to false alarms. Fig. 3 illustrates the BERs of all the reused models, which are all less than the threshold $\tau$ with a maximum of 0.227. The results validate the effectiveness and conferrability of FIT-Print.

## 4.3 Ablation Study

### 4.3.1 Effects of the Length of the Fingerprint

In this experiment, we investigate the impact of varying lengths of the fingerprint $\boldsymbol{F}$. In addition to the default length of $256 = 16 \times 16$, we set the length to be $12 \times 12$, $20 \times 20$, and $24 \times 24$. The results illustrated in Fig. 4 indicate that both FIT-ModelDiff and FIT-LIME can recognize the reused models and the independent models with different lengths of fingerprints. Moreover, with the length of $\boldsymbol{F}$ increases, the BERs of both reused and independent models are more concentrated, signifying that a larger fingerprint length can reduce the probability of outliers and have better security.

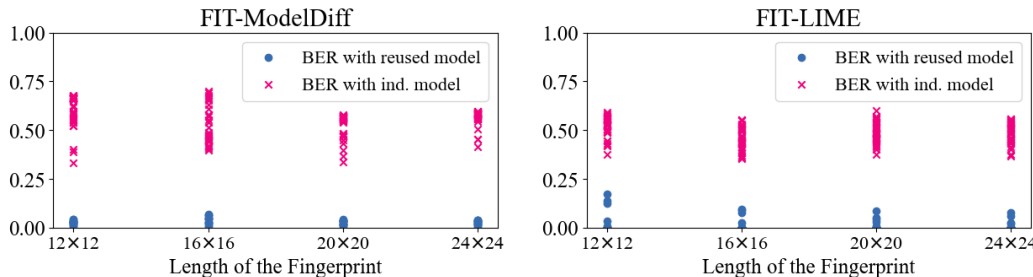

Figure 4: The BERs of the reused models and independent models with different lengths of fingerprint. As the length increases, the BERs with reused and independent models become more concentrated.

Table 3: The average distances of reused models (Avg. Reused Model Dist.) and independent models (Avg. Ind. Model Dist.) and the $\ell_2$-norm of the perturbations $\mathcal{R}$ ($\ell_2$-norm of Pert.) with different $\lambda$. As the $\ell_2$-norm coefficient $\lambda$ increases, the amplitude of perturbations diminishes.

| Method→ | FIT-ModelDiff | | | | | FIT-LIME | | | | |
|---|---|---|---|---|---|---|---|---|---|---|
| Metric ↓ $\lambda$ → | 0.0 | 1.0 | 5.0 | 10.0 | 100.0 | 0.0 | 0.5 | 1.0 | 2.0 | 5.0 |
| Avg. Reused Model Dist. | 0.024 | 0.038 | 0.030 | 0.032 | 0.029 | 0.029 | 0.029 | 0.034 | 0.036 | 0.047 |
| Avg. Ind. Model Dist. | 0.568 | 0.570 | 0.561 | 0.566 | 0.577 | 0.505 | 0.505 | 0.510 | 0.506 | 0.512 |
| $\ell_2$-norm of Pert. | 0.007 | 0.007 | 0.007 | 0.007 | 0.006 | 0.020 | 0.019 | 0.018 | 0.017 | 0.014 |

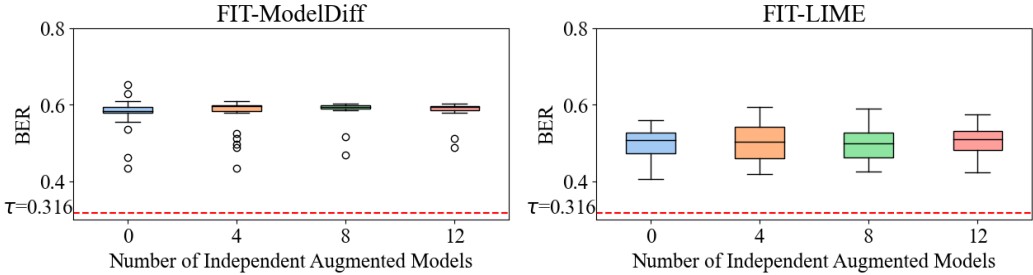

Figure 5: The BERs of the independent models while conducting adaptive false claim attacks using different numbers of independent models as augmented models. The BERs are all larger than the threshold $\tau$, indicating the failure of false claim attacks against FIT-ModelDiff and FIT-LIME.

### 4.3.2 EFFECTS OF THE $\ell_2$-NORM COEFFICIENT

$\lambda$ is the coefficient of the scale of the perturbations in the loss function Eq. (8). In this experiment, we study the effect of $\lambda$ on FIT-Print and adopt FIT-ModelDiff and FIT-LIME with five different $\lambda$. From Table 3, since the scale of the perturbations in FIT-ModelDiff is quite small, varying $\lambda$ does not significantly affect the perturbations as well as the distances with reused models. While in FIT-LIME, a larger $\lambda$ can lead to a smaller perturbation. The $\ell_2$-norm of the perturbations reduces from 0.020 to 0.014. In the meantime, the average distances with reused models become larger. Our experiments also suggest that the effect of $\lambda$ on the distances with independent models is not significant. The visualization of the perturbed testing samples with different $\lambda$ can be found in Appendix I.

### 4.4 THE RESISTANCE TO ADAPTIVE FALSE CLAIM ATTACK

In this subsection, we consider an adaptive false claim attack where the adversary utilizes Eq. (8) to optimize the testing samples, but intentionally crafts some independent models as augmented models to enhance the transferability of the fingerprint. The adversary utilizes the same fingerprint used for its own model to generate transferable samples. We utilize the models pre-trained on ImageNet and their corresponding reused models as the augmented models. The results are shown in Fig. 5. It is demonstrated that adding independent augmented models does not significantly enhance the transferability of the fingerprint in FIT-Print because the BERs on the independent models are nearly unchanged. The results indicate that the adaptive false claim attacks do not work in FIT-Print.

Table 4: The BERs before and after the adaptive overwriting attack and unlearning attack. The BERs after attacks are still low enough to be identified as a reused model. Therefore, our FIT-Print is able to resist the overwriting attack and unlearning attack.

| Method | Before Attack | After Overwriting | After Unlearning |
|---|---|---|---|
| FIT-ModelDiff | 0.047 | 0.051 | 0.149 |
| FIT-LIME | 0.000 | 0.016 | 0.016 |

### 4.5 THE RESISTANCE TO ADAPTIVE FINGERPRINT REMOVAL ATTACKS

In real-world scenarios, the model reuser usually knows which model fingerprinting method is leveraged by the model developer, and can accordingly design an adaptive attack against the utilized model fingerprinting method. Generally speaking, there are two different ways to attack a model fingerprinting method (Yao et al., 2023): **(1)** fine-tuning the model (*i.e.*, model-based attacks) or **(2)** perturbing or preprocessing the input data (*i.e.*, input-based attacks) to obfuscate the fingerprint of the model. In this section, we primarily focus on the former attack. The details and discussions of the latter attack can be found in Appendix D.

In model-based adaptive attacks, the model reuser can fine-tune the model attempting to remove the original fingerprint inside it. Based on the knowledge of the model reuser with the fingerprint of the model developer, we consider two different model-based adaptive attacks.

- *Overwriting Attack:* In overwriting attacks, we assume that the model reuser has no knowledge of the testing samples $\mathcal{X}_T$ and the target fingerprint $\boldsymbol{F}$ utilized by the model developer. Thereby, the model reuser can independently generate the testing samples $\hat{\mathcal{X}}_T$ and the target fingerprint $\hat{\boldsymbol{F}}$, and then fine-tunes the model to make the outputs of $\hat{\mathcal{X}}_T$ close to the target fingerprint $\hat{\boldsymbol{F}}$. The loss function can be defined as follows.

$$\min_{M_o} \frac{1}{|\hat{\mathcal{X}}_T|} \sum_{\hat{\boldsymbol{x}} \in \hat{\mathcal{X}}_T} \mathcal{L}(f(M_o(\hat{\boldsymbol{x}}), \hat{\boldsymbol{F}}). \tag{16}$$

- *Unlearning Attack:* In unlearning attacks, we assume that the model reuser knows the target fingerprint of the model developer, since it may be registered in a third-party institution and publically accessible. However, the model reuser still has no knowledge of the testing samples. As such, the model reuser can construct some independent testing samples to unlearn the target fingerprint $\boldsymbol{F}$ from the model. The loss function can be defined as follows.

$$\max_{M_o} \frac{1}{|\hat{\mathcal{X}}_T|} \sum_{\hat{\boldsymbol{x}} \in \hat{\mathcal{X}}_T} \mathcal{L}(f(M_o(\hat{\boldsymbol{x}}), \boldsymbol{F}). \tag{17}$$

The results of the two adaptive attacks are demonstrated in Table 4. The results indicate that both two attacks cannot successfully remove the fingerprint from the model. Due to more knowledge about the fingerprint $\boldsymbol{F}$, the unlearning attack is slightly more effective than the overwriting attacks but still not able to bypass the ownership verification, with a BER of 0.149. The experimental results show that the model reuser cannot destroy the fingerprint inside the model when having no knowledge of the testing samples. Our FIT-Print can resist both the overwriting attack and the unlearning attack.

## 5 CONCLUSION

In this paper, we revisited existing model fingerprinting methods. We designed a false claim attack by crafting some transferably easy samples and revealed that existing model fingerprinting methods were vulnerable to the false claim attack. We found that the vulnerability can be attributed to the untargeted nature that existing methods compare the outputs of any given samples on different models rather than the similarities to specific signatures. To tackle the above issue, we proposed FIT-Print, a false-claim-resistant model fingerprinting paradigm. FIT-Print transformed the fingerprint of the model into a targeted signature by optimizing the testing samples. We correspondingly designed two fingerprinting methods based on FIT-Print, namely the bit-wise FIT-ModelDiff and the list-wise FIT-LIME. Our empirical experiments demonstrated the effectiveness, conferrability, and resistance to false claim attacks of our FIT-Print. We hope our FIT-Print can provide a new angle on model fingerprinting methods to facilitate more secure and trustworthy model sharing and trading.

## ETHICS STATEMENT

Unauthorized model reuse has posed a serious threat to the intellectual property rights (IPRs) of the model developers. Model fingerprinting is a promising solution to detect reused models. In this paper, we propose a new paradigm of model fingerprinting dubbed FIT-Print. Our FIT-Print is purely defensive and does not discover new threats. Moreover, our work utilizes the open-source dataset and does not infringe on the privacy of any individual. Our work also does not involve any human subject. As such, this work does not raise ethical issues in general.

## REPRODUCIBILITY STATEMENT

The detailed experimental settings of datasets, models, hyper-parameter settings, and computational resources can be found in Section 4.1 and Appendix C.2. The codes and model checkpoints for reproducing our main evaluation results are provided in the supplementary material. We will release the full codes of our methods upon the acceptance of this paper.

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

## APPENDIX

## A   THE DETAILED THREAT MODEL

In this section, we provide a detailed introduction to the threat models of model fingerprinting and false claim attacks. Three parties involved in the threat models are depicted in Figure 6.

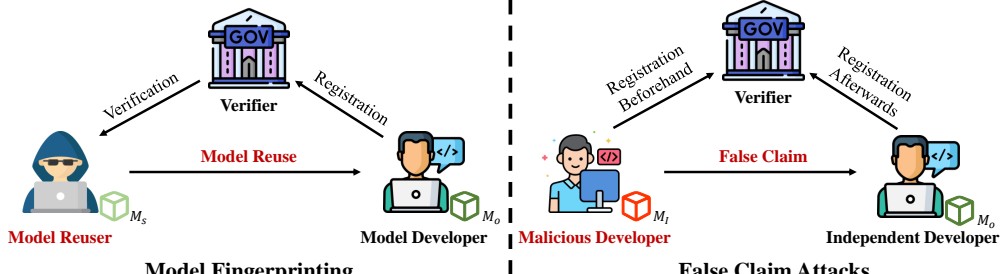

Figure 6: The threat models and detailed processes of model fingerprinting and false claim attacks. In model fingerprinting, the model developer generates and registers the model and the fingerprint to a third-party verifier. Once the model is reused by a model reuser, the verifier can determine the model ownership by comparing the fingerprints. Instead, in false claim attacks, the malicious developer attempts to register a transferable fingerprint to falsely claim other independent developers' models.

## A.1 DETAILED THREAT MODEL OF MODEL FINGERPRINTING

There are three parties involved in the threat model of model fingerprinting, including the model developer, the model reuser, and the verifier. The model developer trains a model and the model reuser attempts to steal and reuse this model. The verifier is responsible for fingerprint registration and ownership verification. The assumptions of these three parties can be found in Section 2.1.

**Process of Model Fingerprinting.** Model fingerprinting can be divided into three steps, including fingerprint generation, fingerprint registration, and ownership verification.

1. **Fingerprint Generation**: In this step, the model developer trains its source model $M_o$ and generates the fingerprint of $M_o$.

2. **Fingerprint Registration**: After generating the fingerprint, the model developer registers the fingerprint and the model with a timestamp to a trustworthy third-party verifier.

3. **Ownership Verification**: For a suspicious model $M_s$ that could be a reused version of $M_o$, the verifier will first check the timestamps of these two models. If the registration timestamp of $M_s$ is later than $M_o$, the verifier will further check whether the fingerprint of $M_o$ is similar to the fingerprint $M_s$. If so, the suspicious model can be regarded as a reused version of $M_o$.

## A.2 DETAILED THREAT MODEL OF FALSE CLAIM ATTACKS

There are three parties involved in the threat model of false claim attacks, including the malicious developer, the verifier, and an independent developer. The formal definition of false claim attacks can be found in Section 2.3.

**Assumption of the Malicious Developer**. In false claim attacks, the malicious developer is the adversary who aims to craft and register a *transferable* fingerprint to falsely claim the ownership of the independent developer's model $M_I$. The malicious developer is assumed to have adequate computational resources and datasets to train a high-performance model and carefully craft transferable model fingerprints. The primary goal of the malicious developer is that the registered model fingerprints can be verified in as many other models as possible. By generating the transferable fingerprint, the malicious developer can (falsely) claim the ownership of any third-party models (that are registered later than that of the malicious developer).

**Process of False Claim Attacks.** The process of false claim attacks can also be divided into three steps, including fingerprint generation, fingerprint registration, and false ownership verification.

1. **Fingerprint Generation**: In this step, the model developer trains its source model $M_o$ and attempts to generate a *transferable* fingerprint of $M_o$.

2. **Fingerprint Registration**: After generating the fingerprint, the model developer registers the *transferable* fingerprint and the model with a timestamp to a trustworthy third-party verifier.

3. **Falsely Ownership Verification**: The adversary tries to use the transferable fingerprint to falsely claim the ownership of another independently trained model $M_I$. Since the fingerprint is registered beforehand, the ownership verification won't be rejected due to the timestamp. Subsequently, the benign developer may be accused of infringement.

## B    THE PROOF OF THEOREM 1

**Theorem 1.** *Given the security parameter $\kappa$ and the fingerprint $\boldsymbol{F} \in \{-1, 1\}^k$, if $\tau$ satisfy that*

$$\sum_{d=0}^{\lfloor \tau k \rfloor} \binom{k}{d}(\frac{1}{2})^k \leq \kappa, \tag{1}$$

*where $\binom{k}{d} = k!/[d!(k-d)!]$, the probability of a successful false claim attack, i.e., the BER is less than $\tau$ with the adversaries testing samples, is less than $\kappa$.*

*Proof.* As mentioned in Section 2.2 and Section 2.3, registering the fingerprint with a timestamp can avoid any afterward false claim attack. As such, the adversary needs to craft the testing samples $\bar{\mathcal{X}}_T$ which can transfered to independent models in advance. We assume that the adversary extracts the fingerprint $\bar{\boldsymbol{F}}$ from the independent model $M_I$ using the testing samples $\bar{\mathcal{X}}_T$, as follows.

$$\bar{\boldsymbol{F}} = \texttt{sign}(M_I(\bar{\mathcal{X}}_T)), \tag{2}$$

We assume that $\check{\boldsymbol{F}}$ denotes the adversary's target fingerprint. Since $\bar{\boldsymbol{F}} \in \{-1, 1\}^k$ is a $k$-bit binary vector and the adversary has no knowledge of the independent model $M_I$, the probability of any bit in $\bar{\boldsymbol{F}}$ to match the corresponding bit in $\check{\boldsymbol{F}}$ is $1/2$. Thus, to satisfy Eq. (5) in Proposition 3, i.e., making the BER between $\bar{\boldsymbol{F}}$ and $\check{\boldsymbol{F}}$ less than $\tau$, there needs to have at least $k - \lfloor \tau \cdot k \rfloor$ bits in $\bar{\boldsymbol{F}}$ match $\check{\boldsymbol{F}}$. Based on the binomial theorem, we have the probability of the aforementioned scenario, i.e., a successful false claim attack, is as follows.

$$\Pr\{\texttt{BER}(\bar{\boldsymbol{F}}, \check{\boldsymbol{F}}) \leq \tau\} = \sum_{d=0}^{\lfloor \tau k \rfloor} \binom{k}{d}(\frac{1}{2})^k. \tag{3}$$

Since the right-hand side of the Eq. (3) is less than $\kappa$, the probability of a successful false claim attack, i.e., the BER is less than $\tau$ with the adversarial testing samples, is also less than $\kappa$.    □

## C    IMPLEMENTATION DETAILS

### C.1    DETAILS OF THE MODEL REUSE TECHNIQUES

In our experiments, we evaluate FIT-Print and other different model fingerprinting methods against the following five categories of model reuse techniques, including copying, fine-tuning, pruning, model extraction, and transfer learning.

- **Copying**: Copying refers to the adversary somehow gaining white-box access to the parameters and architecture of the victim model. Subsequently, the adversary steals the model by directly copying it. It may occur when the model is open-source and publicly available.

- **Fine-tuning**: Fine-tuning means the adversary re-trains the victim model with its own dataset which is related to the primitive task of the victim model. We consider three types of fine-tuning denoted as `Fine-tuning(10%)`, `Fine-tuning(50%)`, and `Fine-tuning(100%)`, which means we fine-tune the last 10%, 50%, and 100% layers of the victim models.

- **Pruning**: Pruning intends to compress the model by removing redundant parameters. We leverage weight pruning (Han et al., 2015) as our pruning method, which prunes the neurons according to their activations. We prune 10%, 30%, and 50% parameters of the victim model, denoted as `Pruning(10%)`, `Pruning(30%)`, and `Pruning(50%)` respectively.

- **Model Extraction**: Model extraction attempts to steal the knowledge of the victim model via only black-box access. The adversary can obtain the output of the victim model to train the extracted model. In our experiments, we implement the model extraction in two different scenarios. `Extract(same)` and `Extract(different)` refer to utilizing the same or different model architectures to extract the source model, respectively.

- **Transfer Learning**: Transfer learning is an ML technique where the victim model trained on one task is adapted as the starting point for a model on the second related task. In our experiments, we replace the last layer of the model to fit the second task and fine-tune the model for 200 epochs.

Similar to the setting of fine-tuning, we fine-tune the last 10%, 50%, and 100% layers of the victim models to implement transfer learning. These models are denoted as `Transfer(10%)`, `Transfer(50%)`, and `Transfer(100%)` respectively.

## C.2    Details of the Experimental Settings

In this section, we introduce the details of the experimental settings, including the optimization details, details of datasets, and computational resources.

**Optimization Details.** We utilize the stochastic gradient descent (SGD) with momentum as the optimizer. We set the initial learning rate to $1.2 \times 10^{-2}$, the momentum to $0.9$, and the weight decay to $5 \times 10^{-4}$. We apply a cosine annealing schedule (Loshchilov & Hutter, 2016) to reduce the learning rate gradually to a minimum of $4 \times 10^{-3}$. Following (Shao et al., 2024a), we set the control parameter $\varepsilon$ in the hinge-like loss in Eq. (7) to $0.01$. We optimize the perturbations on the testing samples for 300 epochs. In Eq. (13) of FIT-ModelDiff, we set the bias parameter $\alpha$ to be $7\pi/8$.

**Details of Datasets.** In this paper, we mainly utilize three different datasets, including Flowers102, SDogs120, and ImageNet. Flowers102 is an image classification dataset consisting of 102 categories of flowers. Each class consists of between 40 and 258 images. SDogs120 dataset contains images of 120 breeds of dogs from around the world. This dataset has been built using images and annotation from ImageNet for the task of fine-grained image categorization. SDogs120 includes 20,580 images in total. ImageNet dataset is a large image database that has images from 1,000 different classes. In our experiments, we utilize the images from ImageNet as the initial values of the testing samples. For simplicity, all the images used in our experiments are resized to $224 \times 224 \times 3$.

**Computational Resources.** In our implementations, we utilize PyTorch as the deep learning framework. All our experiments are implemented with 8 RTX 3090 GPUs.

## D    The Resistance to Input-based Adaptive Attacks

In this type of adaptive attack, the model reuser can deliberately perturb or preprocess the input data and make the output of any input data away from the target fingerprint $\boldsymbol{F}$. Since the model reuser does not know which input is one of the testing samples, it has to perturb all the inputs to bypass the ownership verification. Although this type of attack may prevent extracting the correct fingerprint from the suspicious model, we argue that FIT-Print is still practical for the following two reasons.

- Input-based adaptive attack is extremely costly to implement. Perturbing or preprocessing all the input samples may require enormous computational resources.

- Input-based adaptive attack compromises the functionality of the model. The model reuser also needs to perturb benign samples, leading to a degradation of the utility of the model.

Table 5: The false positive rates (FPR) of existing model fingerprinting methods and our FIT-ModelDiff and FIT-LIME before and after false positive attacks.

| Metric↓ Method→ | IPGurad | ModelDiff | Zest | SAC | FIT-ModelDiff | FIT-LIME |
|:---:|:---:|:---:|:---:|:---:|:---:|:---:|
| FPR | 30.6% | 4.0% | 7.6% | 39.6% | 0.0% | 0.0% |
| FPR After Attacks | 61.8% | 15.28% | 29.51% | 51.94% | 0.0% | 0.0% |

## E    The Omitted Results of False Claim Attacks

In this section, we present the full results of false claim attacks against existing model fingerprinting methods and our FIT-ModelDiff and FIT-LIME. Specifically,

- For AE-based methods, designing a false claim attack is equivalent to designing a transferable adversarial attack. We utilize the false claim attack proposed in Liu et al. (2024).

- For testing-based methods, we design a false claim attack in Section 2.3.

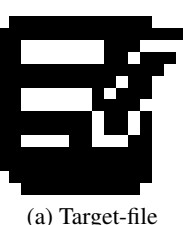
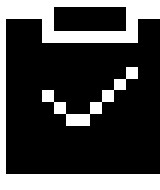
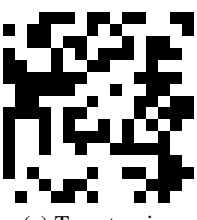

(a) Target-file           (b) Target-tick           (c) Target-noise

Figure 7: The visualization of the targeted fingerprints. We utilize three different target fingerprints in our experiments. The 'Target-file' fingerprint is used in our main experiments.

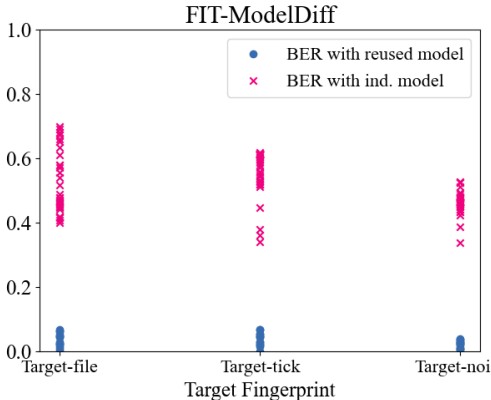
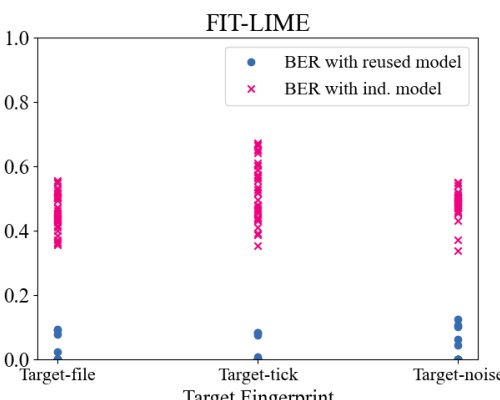

Figure 8: The BERs of the reused models and independent models with different target fingerprints. Regardless of the target fingerprints, our proposed FIT-ModelDiff and FIT-LIME have the capability to distinguish the reused models and the independent models.

The results in Table 5 show that false claim attacks are effective against existing fingerprinting methods and demonstrate that performing false claim attacks can significantly increase FPR.

## F  ADDITIONAL ABLATION STUDY

### F.1  FIT-PRINT WITH DIFFERENT TARGET FINGERPRINTS

**How to Choose the Target Fingerprint $F$.** We briefly introduce how to choose the target fingerprint. In our method, the targeted fingerprint is a bit string representing the identity of the model developer and needs to be registered to the trustworthy verifier. For instance, the company's logo or personal identity number can be used as a targeted fingerprint. We note that the choice of the fingerprint does not affect the model performance. This is because model fingerprinting does not alter the models' parameters and has no impact on the model performance. This is a key advantage of fingerprinting.

**Experiments with Different Target Fingerprints.** In the main experiments of our paper, we utilize an image of a file and a pen as the target fingerprint $F$. In this section, we explore the use of different images as the target fingerprint and validate the effectiveness of FIT-Print regardless of the target fingerprints. Specifically, we choose two target fingerprints, one is a tick image and the other is the random noise. The visualization of the three fingerprints (resized to $16 \times 16$ bits) is shown in Fig. 7.

The experimental results are shown in Fig. 8. From Fig. 8, we can find that regardless of the target fingerprints, all the BERs of reused models are lower and the BERs of independent models are larger than the threshold. This demonstrates that our FIT-Print is effective with different target fingerprints.

### F.2  FIT-PRINT WITH DIFFERENT INITIALIZATIONS OF TESTING SAMPLES

In this section, we evaluate whether the initialization of the testing samples influences the effectiveness of FIT-Print. Drawing inspiration from the design of trigger samples in model watermarking methods,

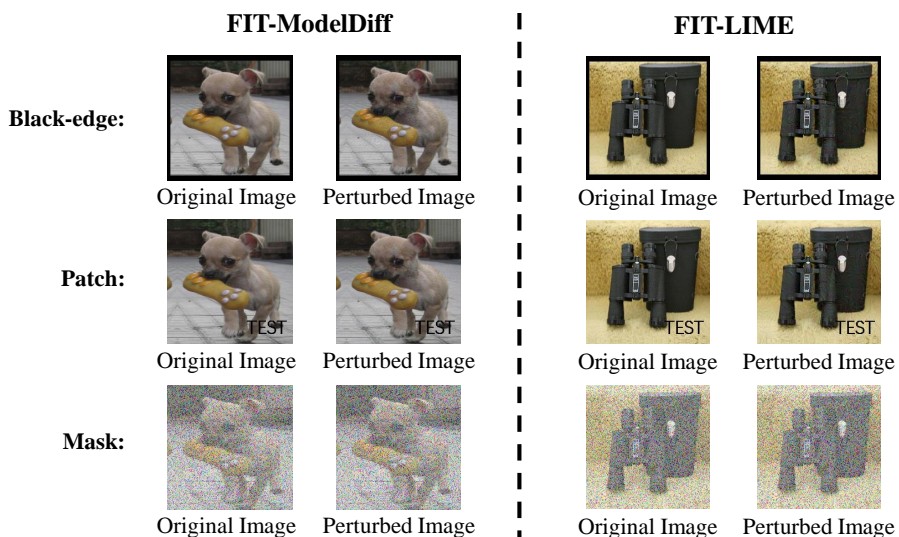

Figure 9: The visualization of the original images and perturbed images with different initializations.

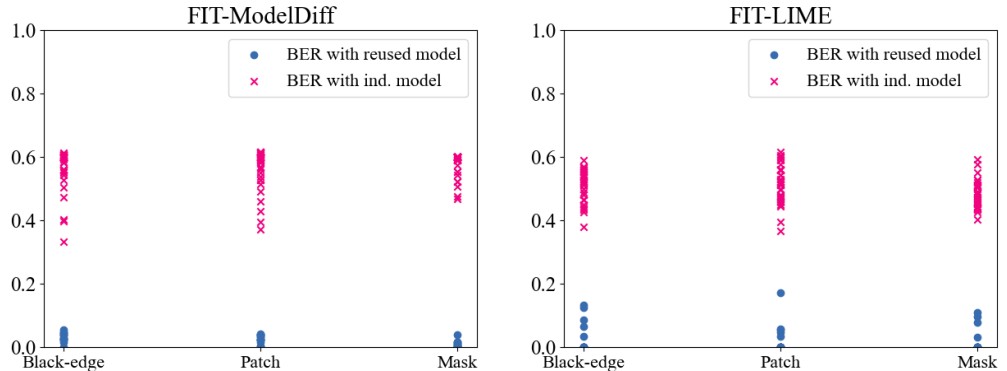

Figure 10: The BERs of the reused models and the independent models with different initializations. Our proposed FIT-Print can work well no matter which initialization method is used.

we consider the following three testing sample initialization methods, denoted as 'Black-edge' (Shao et al., 2024a), 'Patch' (Zhang et al., 2018), and 'Mask' (Guo & Potkonjak, 2018), respectively.

- **Black-edge**: 'Black-edge' first randomly selects the benign images from the dataset and adds a black edge around the images. We leverage this initialization method in our main experiments.

- **Patch**: 'Patch' sticks some meaningful patch (*e.g.*, 'TEST' or any pattern representing the identity of the model developer) into the images.

- **Mask**: 'Mask' adds noise to the images. The noise is pseudo-random and the seed to generate the noise is associated with the identity of the model developer.

The visualization of the four initialization methods and their perturbed version are shown in Fig. 9 and the experimental results are shown in Fig. 10. The results demonstrate that FIT-Print successfully distinguishes the reused models and the independent models since all the BERs of independent models are larger than $\tau$ and all the BERs of reused models are less than $\tau$. The results indicate the effectiveness of FIT-Print regardless of the initialization methods.

## F.3 FIT-PRINT WITH DIFFERENT NUMBERS OF AUGMENTED MODELS

In the testing sample extraction stage, FIT-Print utilizes the reused models as augmented models to enhance the conferrability of the fingerprint. In this section, we study to leverage different numbers of

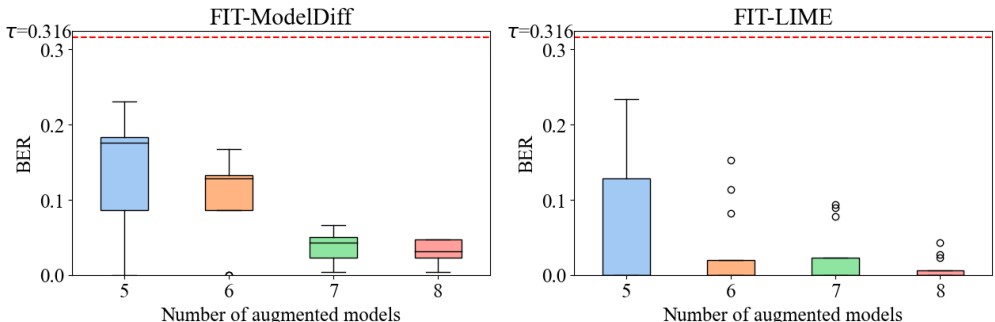

Figure 11: The BERs of the reused models with different numbers of augmented models. Using more models for augmentations can have lower BERs.

Table 6: The performance of FIT-ModelDiff and FIT-LIME in the label-only scenario. The results demonstrate that FIT-Print can still distinguish the reused models with only top-1 labels.

| Metric↓ Method→ | FIT-ModelDiff | FIT-LIME |
|---|---|---|
| Avg. BER | 0.227 | 0.135 |
| Ownership Verification Rate | 1.000 | 1.000 |
| Avg. BER of Ind. Models | 0.369 | 0.399 |
| False Positive Rate | 0.000 | 0.000 |

augmented models to extract the testing samples and test whether FIT-Print maintains a satisfactory conferrability. Fig. 11 depicts the BERs of the reused models with different numbers of augmented models. We set the number to 5, 6, 7, 8. From Fig. 11, we can find that as the number of augmented models increases, the BERs on reused models become smaller and more concentrated, which means using more reused models as augmented models can enhance the conferability of FIT-Print. In addition, while using only 5 reused models as augmented models, all the BERs are smaller than the threshold $\tau$, which signifies the conferrability of FIT-Print.

## G  FIT-PRINT IN THE LABEL-ONLY SCENARIO

In this section, we investigate the effectiveness of FIT-Print in the label-only scenario. In such a scenario, the verifier can only obtain the Top-1 label instead of logits as output. Unfortunately, detecting transfer learning models, which is one of our considered important model reuse settings, is still an open problem in model ownership verification (Sun et al., 2023). Transfer learning can change the task of the model and the output classes. It is hard to determine whether a model is transferred from another with only top-1 labels. Consequently, in the following discussion, we do not take transfer learning models into account.

FIT-Print can easily be extended to distinguish the reused models (except transfer learning models) with the top-1 labels. In the label-only scenario, we can construct a binary vector $b$ to replace the original logits. Assuming that the predicted top-1 class is $a$, the $a$-th element in $b$ is set to 1 and the other elements are 0. The other processes remain unchanged.

To verify the effectiveness, we conduct additional experiments. Table 6 shows the average bit error rate (Avg. BER) on the reused models and the independent models (Ind. Models). We also present the ownership verification rates on the reused models and the false positive rates on the independent models. The results show that our methods are still highly effective under the label-only setting, although the average BERs of the reused models decrease in this scenario.

## H  THE OVERHEAD OF FIT-MODELDIFF AND FIT-LIME

Compared with existing model fingerprinting methods, FIT-Print needs to optimize the testing samples and thus has an extra overhead. We hereby present a detailed analysis of the time and space complexity of the two fingerprinting methods, FIT-ModelDiff and FIT-LIME. FIT-ModelDiff and

FIT-LIME are the representatives of bit-wise and list-wise methods, respectively. As such, they have different trade-offs in time and space overhead.

**During the fingerprint verification stage:**

- For FIT-ModelDiff, the space complexity is $O(1)$ and the time complexity is $O(k)$ where k is the length of the targeted fingerprint. FIT-ModelDiff is a bit-wise fingerprinting method that extracts the fingerprint bit by bit. It only reads two samples to calculate one bit in the fingerprint in each step. The fingerprint can be extracted in k steps. Therefore, the space complexity is $O(1)$, and the time complexity is $O(k)$.

- For FIT-LIME, the space complexity is $O(k)$ and the time complexity is $O(k/\beta)$ where $\beta$ is the batch size. FIT-LIME is a list-wise method that extracts the fingerprint as a whole list. FIT-LIME needs to read all the samples at the same time to extract the fingerprint. On the other hand, FIT-LIME can calculate the outputs of an entire batch at the same time. As such, the space complexity is $O(k)$, and the time complexity is $O(k/\beta)$.

**During the testing samples extraction stage:**

In each iteration of optimizing the testing samples, we need to perform one forward propagation and one backward propagation of the fingerprint verification method. Assuming that we utilize $\xi$ augmented models during optimization, the time complexities of each iteration of testing sample extraction in FIT-ModelDiff and FIT-LIME are $O(\xi \cdot k)$ and $O(\xi \cdot k/\beta)$**. $k$ is the length of the targeted fingerprint and $\beta$ is the batch size. For instance, in our main experiments, we utilize 10 augmented models and a 256-bit targeted fingerprint. It takes nearly 3 seconds for one optimization iteration in FIT-ModelDiff and 1 second for that in FIT-LIME. As such, our FIT-ModelDiff and FIT-LIME are efficient in optimization and the overhead is acceptable.

## I  THE VISUALIZATION OF THE TESTING SAMPLES

In this section, we present the visualization of the extracted testing samples with different $\lambda$. As shown in Figure 12, for both FIT-ModelDiff and FIT-LIME, the perturbations on the testing samples are nearly visually imperceptible. Moreover, according to our quantitative experiments in Section 4.3.2, a larger $\lambda$ can regulate the magnitude of the perturbation and thus lead to a smaller perturbation. This conclusion can also be confirmed in Figure 12.

## J  EXTENDING FIT-PRINT TO OTHER MODELS AND DATASETS

In our main experiments, we focus on image classification models and datasets. It is also technically feasible to extend our FIT-Print to models of any task. In this section, we discuss how FIT-Print can generalize to different types of models and data.

### J.1  THE EXTENSION TO OTHER MODELS

We argue that our FIT-Print can generalize to models with different architectures and tasks. For models with different architectures, since we do not make any assumptions about the architecture of the models and we also do not need to alter or fine-tune the model, our method can fundamentally generalize to models with other architectures (*e.g.*, transformers) as well. For models with different tasks, the major difference between models with different tasks is the output format. For instance, the image generation model outputs a tensor consisting of a sequence of logits. FIT-ModelDiff calculates the cosine similarity between the outputs and FIT-LIME calculates the average entropy of the output. The two calculation methods can be applied to any output format (*e.g.*, 1-D vectors, 2-D matrices, or tensors). As such, our methods are naturally feasible for models with different tasks.

### J.2  THE EXTENSION TO OTHER (TYPES OF) DATASETS

Our FIT-Print can also generalize to other types of datasets. Our primitive FIT-Print aims to optimize a perturbation $\boldsymbol{r}$ on the input $\boldsymbol{x}$ to make the mapping vector close to the targeted fingerprint. The main part of the loss function is as Eq. (4).

$$\min \mathcal{L}(f(M_o(\boldsymbol{x} + \boldsymbol{r}), \boldsymbol{F}).$$  (4)

**FIT-ModelDiff**

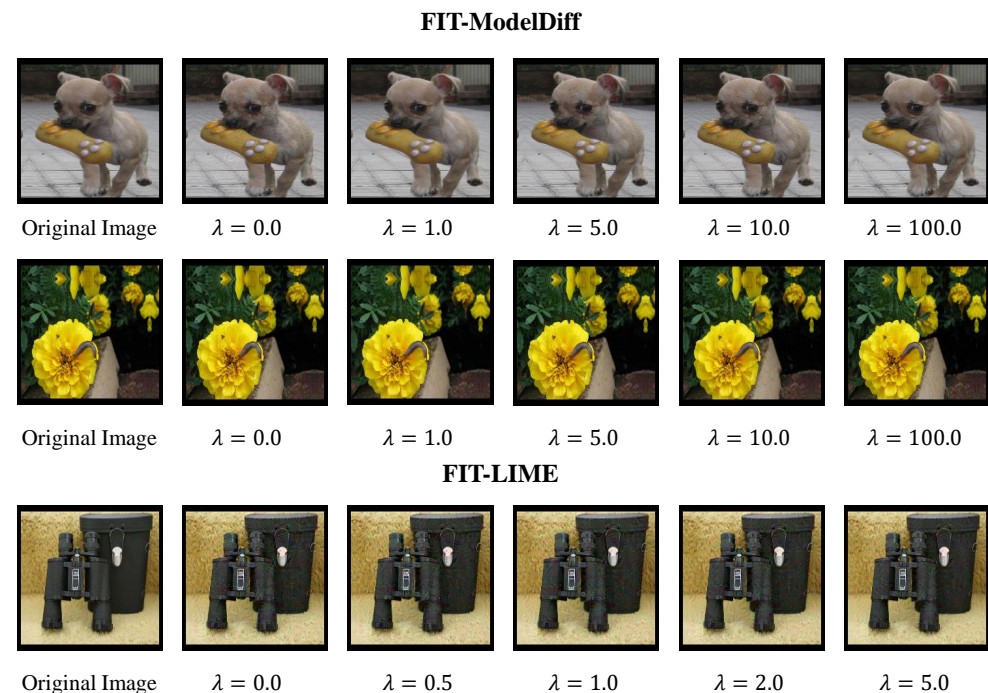

Figure 12: The visualization of the original images and the perturbed testing samples with different $\lambda$.

However, for the discrete data (*e.g.*, text data), it is not feasible to directly add the perturbation to it. Thus, a rewriting function $g(\boldsymbol{x})$ can be introduced to rewrite the characters, words, or sentences. The loss function can be changed to Eq. (5).

$$\min \mathcal{L}(f(M_o(g(\boldsymbol{x})), \boldsymbol{F}). \tag{5}$$

Arguably, the main challenge lies in how to design an effective optimization method to find a rewriting function $g(\boldsymbol{x})$ which minimizes the above loss function. There are already some existing works (Guo et al., 2021; Yao et al., 2024; Wen et al., 2024) to fulfill this task. Accordingly, our FIT-Print can be adapted to other data formats (*e.g.*, text or tabular).

### J.3 CASE STUDY ON TEXT GENERATION MODEL

In this section, we conduct a case study on implementing FIT-ModelDiff and FIT-LIME on text generation models. Text generation models (OpenAI, 2023) have become the most famous models in recent years and have been widely applied in various domains. Specifically, the text generation model predicts the next token in a sequence of tokens, *i.e.*, the output of the text generation models is a sequence of logits. Given an input sequence $\boldsymbol{s} = \{s_1, s_2, ..., s_q\}$, where $q$ is the number of tokens in the sequence, and a vocabulary $\mathcal{V}$, the text generation model outputs a sequence $\boldsymbol{o} \in \mathbb{R}^{q \times |\mathcal{V}|}$. The $i$-th element in $\boldsymbol{o}$ is the probability logit of the tokens in the vocabulary.

To implement FIT-Print on text generation models, we need to optimize Eq. (5) to generate the testing samples. Arguably, our FIT-ModelDiff and FIT-LIME can easily generalize to protect text generation models. Specifically, the mapping functions used in FIT-ModelDiff and FIT-LIME (*i.e.*, cosine similarity and average entropy) can be directly applied to text generation models. This is because text generation models differ from classification models only in the output dimension and these two functions are inherently able to calculate data with different dimensions. The main challenge is to optimize the discrete text data to minimize Eq. (5). To achieve this goal, we can exploit existing text optimization methods (Guo et al., 2021; Wen et al., 2024). Specifically, we implement the optimization method proposed in (Wen et al., 2024). It optimizes the embeddings of the text sequences and then finds the nearest token in the embedding space to replace the original token.

We further conduct experiments to verify the effectiveness of our FIT-ModelDiff and FIT-LIME on text generation models. We use two popular text generation models (*i.e.*, GPT-2 (Radford et al.,

Table 7: The bit error rates (BER) of applying FIT-ModelDiff and FIT-LIME to text generation models (lower is better).

| Method→ | FIT-ModelDiff | | FIT-LIME | |
|---|---|---|---|---|
| Dataset↓ Model→ | GPT-2 | BERT | GPT-2 | BERT |
| ptb-text-only | 0.188 | 0.188 | 0.031 | 0.000 |
| lambada | 0.188 | 0.125 | 0.062 | 0.000 |

2019) and BERT (Devlin et al., 2018)) and two datasets (*i.e.*, ptb-text-only (Marcus et al., 1993) and lambada (Paperno et al., 2016)) for our case study. Table 7 shows the bit error rates (BERs) of applying our methods to text generation models. The BERs are all lower than the threshold $\tau = 0.227$, indicating that FIT-Print is also applicable to protect the IPR on other data formats.

## K    DISCUSSION ON THE GENERALIZATION OF FIT-PRINT

In this section, we discuss the generalization of our proposed FIT-Print.

**How to Transform Existing Fingerprinting methods into the FIT-Print Paradigm.** In Section 3.5, we propose two targeted model fingerprinting methods as the representatives of bit-wise fingerprinting and list-wise fingerprinting. Based on the insight of these two methods, any existing testing-based model fingerprinting methods can be transformed into the FIT-Print paradigm within two steps.

- First, for the bit-wise fingerprinting methods that extract the fingerprint bit by bit, we need to formulate a sort or location mapping rule to confirm the position of each bit in the fingerprint $\boldsymbol{F}$. The rule can ensure the fingerprint of the model is uniquely determined. The list-wise fingerprinting methods are not necessitated to do so since the fingerprint is extracted as a whole, and the position of the bits in the fingerprint is already determined.

- Second, we need to transform the value range of each element in the fingerprint vector $\boldsymbol{v}$ into an interval containing both positive and negative values by a linear transformation.

After the above two steps, we can utilize the transformed mapping function to develop a new FIT-Print model fingerprinting method and leverage the procedures introduced in Section 3.3 and Section 3.4 to extract the testing samples and ownership verification.

**The Design Criteria for a Mapping Function.** The design criteria for a good mapping function are from the following four main aspects.

- **Distinguishable:** Different models need to exhibit different outputs in the output space of the mapping function. This can guarantee that applying the mapping function can distinguish different independent models.

- **Task-agnostic:** The mapping function needs to be able to process the outputs of models with different tasks (*e.g.*, with different number of classes).

- **Robust:** The outputs of the mapping function on a model need to be robust against various model reusing techniques, *i.e.*, the outputs do not change significantly after model reusing.

- **Efficient**: The calculation of the mapping function needs to be efficient and take a small overhead.

## L    RELATED WORK

### L.1    MODEL WATERMARKING

Model watermarking methods aim to embed an owner-specific signature (*i.e.*, watermark) into the models. In case the watermarked model is reused or stolen by the adversary, the model developer can extract the watermark inside the adversary's suspicious model. If the extracted watermark is similar to the watermark of the model developer, the model developer can accuse the adversary of infringement. Broadly, model watermarking methods can be divided into two categories, white-box model watermarking and black-box model watermarking (Sun et al., 2023; Shao et al., 2024b).

**White-box Model Watermarking Methods**: White-box model watermarking methods assume that the model developer can get full access to the suspicious model during ownership verification. It usually occurs when the adversary publishes the model as an open-source model. White-box model watermarking methods directly embed the watermark into the parameters of the models. For instance, Uchida et.al (Uchida et al., 2017) proposed to add a watermark regularization term into the loss function during fine-tuning to embed the watermark. Darvish et.al (Darvish Rouhani et al., 2019) chose to embed the watermark into the intermediate outputs of the protected models. The watermark can also be embedded into the model by adjusting the architecture of the models (Fan et al., 2019; Lv et al., 2023) or embedding external features (Li et al., 2022c). White-box model watermarking methods need to know the parameters of the suspicious models during ownership verification. This assumption is difficult to realize in practical scenarios because the model is usually deployed in the cloud and can only be accessed via API. Such a limitation restricts the application of the white-box model watermarking methods in the real world.

**Black-box model watermarking methods**: Black-box model watermarking methods assume that the model developer can only observe the outputs from the suspicious models (Adi et al., 2018; Wei et al., 2024). Due to such a constraint, black-box methods are primarily based on backdoor attacks (Li et al., 2022b; Xiang et al., 2023; Gao et al., 2020). Backdoor-based model watermarking methods leverage backdoor attacks to force the model to remember specific patterns and their corresponding target labels (Ma et al., 2023). For ownership verification, the model developer can embed a specific dataset in which each data has a wrong label as watermarks into the model. The trigger set is unique to the watermarked model. The model developer can trigger the misclassification to verify its ownership. Backdoor-based methods can apply to various tasks, such as image classification (Adi et al., 2018; Li et al., 2019), image processing (Zhang et al., 2020a; 2021), federated learning (Liu et al., 2021; Yu et al., 2023), and prompt (Yao et al., 2024). For non-backdoor black-box methods, Miani et al. (Maini et al., 2020) proposed Dataset Inference to implement ownership verification. Recently, Shao et al. (Shao et al., 2024a) proposed embedding a multi-bit watermark into the feature attribution explanations of some specific samples, which can tackle the harmfulness and ambiguity of the backdoor-based model watermarking methods.

However, since model watermarking methods need to embed the watermark into the model through fine-tuning, they inevitably have a negative impact on the functionality of the protected models. Model watermarking methods might reduce the practical value of the models. In addition, as the parameter scale of the model gets larger (*e.g.*, large foundation models (Chang et al., 2024)), it is more costly for the model developer to fine-tune the models, limiting the practical application of model watermarking methods in real world.

### L.2 Model Fingerprinting or Model Functional Distance

In this section, we provide a comprehensive discussion on model fingerprinting. Some existing studies have been developed to compute the functional distance between different models. Although these works serve a different purpose from model fingerprinting for ownership verification, they share technical similarities. Therefore, we collectively refer to these works as model fingerprinting. Similar to model watermarking methods, model fingerprinting methods can also be categorized into white-box and black-box (Sun et al., 2023).

**White-box Model Fingerprinting Methods**: in the white-box scenario, since the model developer can get access to the parameters of the suspicious model, a direct way to compare the models is to compare the weights (or their hash values) of the models. Some existing white-box model fingerprinting methods leveraged the path of model training (Jia et al., 2021), the random projection of model weights (Zheng et al., 2022), or the learnable hash of the model weights (Xiong et al., 2022). Some recent works also explored utilizing the deep representations (*e.g.*, gradients (Song et al., 2023)) or the intermediate results (Chen et al., 2022a) of the testing samples as the fingerprint. However, similar to white-box model watermarking methods, the application of white-box fingerprinting methods in real-world scenarios is also limited.

**Black-box Model Fingerprinting Methods**: in the black-box scenario, the model developer is assumed to have only API access to the suspicious model. Existing black-box model fingerprinting methods can be classified into adversarial-example-based (AE-based) methods (Cao et al., 2021; Wang et al., 2021; Lukas et al., 2021) and testing-based methods (Li et al., 2021; Chen et al., 2022b;a)

and we have presented their formulations in Section 2.2. AE-based methods craft adversarial examples to identify the decision boundary of different models (Cao et al., 2021). Lukas et.al (Lukas et al., 2021) proposed to craft some reused models as augmented models to improve the conferrability of the fingerprint. On the contrary, testing-based methods compare the model behavior on the testing samples at the specific mapping function. ModelDiff (Li et al., 2021) and SAC (Guan et al., 2022) utilized the distances between the output logits of different input samples, while Zest (Jia et al., 2022) took the feature attribution map output by LIME (Ribeiro et al., 2016) as the mapping function. In addition, Chen et.al (Chen et al., 2022a) proposed a series of mapping functions to calculate model similarity. Compared to AE-based methods, testing-based methods have the capability to compare models across different tasks and output formats, thereby attracting greater attention.

There are also some existing works exploring other applications of model fingerprinting. For instance, He et al. (2019) attempts to verify whether the model parameters are altered by the adversary. Specifically, He et al. (2019) aims to generate a fragile fingerprint that can be destroyed when the model is modified by others.

Unlike model watermarking methods, model fingerprinting does not need to alter the parameters, architectures, and training processes of the models. The efficiency of the extraction and verification of model fingerprinting methods is also much higher than model watermarking methods. As such, currently model fingerprinting may be a promising way to protect the IPR of the valuable models.

## M    THE COMPARISON TO SIMILAR WORKS

### M.1    THE COMPARISON OF FALSE CLAIM ATTACK TO OTHER WORKS

**Differences between False Claim Attack and Ambiguity Attack.**  Similar to the false claim attack (Liu et al., 2024), ambiguity attack (Fan et al., 2019) is another attack attempting to forge the ownership certificate and falsely claim to have ownership of another party's model. The major difference between these two attacks is that the ambiguity attack is conducted on a given trained model while the false claim attack aims to create a transferable certificate to claim the ownership of the third-party models trained afterward. Existing literature Liu et al. (2024); Waheed et al. (2024) demonstrates that the registration of ownership certificates ($e.g.$, watermarks or fingerprints) can effectively mitigate ambiguity attacks but is not effective in defending against the false claim attack. As such, the false claim attack can be considered as an improved version of the ambiguity attack and we mainly focus on the false claim attack in this paper.

**Differences between False Claim Attack and False Positive Rate.**  Some existing works (Li et al., 2021; Chen et al., 2022a) may involve the false positive rate, which evaluates whether a fingerprint can be extracted or verified on an independent model (instead of the reused model). The major difference is that the false claim attack is designed to maliciously generate a transferable fingerprint. Contrarily, when calculating the false positive rate, the fingerprint is extracted innocently. Arguably, achieving resistance to the false claim attack is more difficult than a low false positive rate. We also present the false positive rates of FIT-Print in Table 2.

### M.2    THE COMPARISON OF FIT-PRINT TO EXISTING FINGERPRINTING METHODS

**Connections and Differences with ModelDiff:**  The insight of ModelDiff and FIT-ModelDiff is to compare the output differences between perturbed samples and original samples. However, primitive ModelDiff calculated the cosine similarity of these outputs as the similarity score. Since the value range of cosine similarity with positive vectors is always larger than 0, ModelDiff cannot be directly applied to FIT-Print. Also, ModelDiff cannot recognize the models extracted from the source model. FIT-ModelDiff tackled these issues by designing a value range transformation and leveraging augmented models to improve conferrability. The details can be found in Section 3.5.1.

**Connections and Differences with Zest:**  In our FIT-LIME, we exploit the feature attribution output by LIME as the fingerprint. An existing fingerprinting method, Zest (Jia et al., 2022), utilizes primitive LIME to compare different models. However, primitive LIME has two drawbacks in ownership verification: **(1)** LIME first clusters the pixels into several groups called superpixels via Quickshift (Vedaldi & Soatto, 2008). The clustering algorithm is time-consuming and these superpixels are irregular and unordered, making it hard to transform them into a bit string that

represents the fingerprint. **(2)** Primitive LIME depends on the label of the input to calculate the importance scores, which is not applicable when the suspicious model has different predicted classes from the source model. Our proposed FIT-LIME has tackled the above issues, and the technical details can be found in Section 3.5.2.

**Connections and Differences with MetaV:** MetaV(Pan et al., 2022) introduces two critical components, the adaptive fingerprint and the meta-verifier. The adaptive fingerprint is a set of adversarial examples. The meta-verifier takes the suspicious model's output of the adaptive fingerprint and outputs whether the suspicious model is reused from the original model. MetaV accomplishes such an objective by simultaneously optimizing the adaptive fingerprint (*i.e.*, adversarial perturbations) and the meta-verifier (*i.e.*, a fully-connected neural network). In conclusion, MetaV provided a task-agnostic fingerprinting framework. However, MetaV is vulnerable to false claim attacks. MetaV is an AE-based fingerprinting method and the adversary can craft transferable adversarial examples to achieve false claim attacks. This proposition is also presented in Liu et al. (2024). Moreover, MetaV cannot detect transfer learning models, which is one of the realistic stealing settings. MetaV depends on a pre-trained meta-verifier. Transfer learning models may have different output formats, *e.g.*, the number of classes. Therefore, the meta-verifier which has a fixed input format is not able to process the changed outputs of the suspicious model and detect whether it is reused from the original model.

## N   POTENTIAL SOCIETAL IMPACT

In terms of positive societal impact, this paper aims to address the challenges associated with false claim attacks in ownership verification through the utilization of targeted model fingerprinting methods. Our FIT-Print, as a method for protecting intellectual property rights (IPR) related to models, will assist both academia and industry in safeguarding the costly models' IPRs and preventing unauthorized model reuse and theft. Furthermore, FIT-Print has the potential to facilitate the emergence of new business models such as model trading.

On the other hand, one potential negative societal impact is that the insight of FIT-Print is to some extent similar to those of targeted adversarial attacks. Therefore, the insight of FIT-Print might also apply to adversarial attacks. However, FIT-ModelDiff changes the difference between the outputs and FIT-LIME changes the explanation. Neither of them directly turns the prediction classes into a target class. As such, although the insight might be transferred to adversarial attacks, the negative impact of this attack is negligible to most of the AI applications.

## O   POTENTIAL LIMITATIONS AND FUTURE DIRECTIONS

One potential limitation of our FIT-Print is that FIT-Print depends on a trustworthy third-party institution to register the fingerprint with a timestamp. However, we argue that the existing intellectual property office (IPO) or artificial intelligence regulator (AIR) can be responsible for this duty. First, it is common for developers to register their intellectual property, including valuable models, with the IPO for copyright protection. Second, many countries and regions are in the process of establishing or have established the AIR (*e.g.*, as exemplified in the EU Artificial Intelligence Act) to ensure security and transparency before deploying the AI models. As such, it is also feasible for the AIR to manage model registrations and ensure that the new models do not infringe on other's legal copyrights.

Another potential limitation is that FIT-Print does not provide formal proof of the resistance to false claim attacks. As such, a more powerful adversary may still be able to conduct a successful false claim attack. We will investigate how to achieve a certified robust model fingerprinting method against false claim attacks in our future work.

Furthermore, as discussed in Appendix J, for deterministic models that do not involve randomness (*e.g.*, CNN and LLM), our FIT-Print can generalize to different types of models and datasets. However, for non-deterministic models that involve randomness (*e.g.*, diffusion models), we have to admit that we do not know whether our methods and existing fingerprinting methods can be adapted to them since they have a completely different inference paradigm. We will conduct a comprehensive study in our future work.

## P    DISCUSSION ON RESULTS OF THE BASELINE FINGERPRINTING METHODS

In our main experiments, we take five different model fingerprinting methods, IPGuard (Cao et al., 2021), ModelDiff (Li et al., 2021), Zest (Jia et al., 2022), SAC (Guan et al., 2022), and Model-GiF (Song et al., 2023), as our baseline methods. Except for IPGuard, we all use the open-source code provided in the papers for the experiments. These methods work well in their benchmarks but for the sake of fairness, we propose a new benchmark of model reuse and test them in our benchmark. Our benchmark utilizes two datasets (Flowers102 and SDogs120) with a large number of classes (compared with Cifar-10 (Krizhevsky et al., 2009) in SAC) and we also take the independently trained models which have the same task with the source model into account. As such, some existing methods may not work well in our benchmark. For instance, in Table 1, SAC fails to identify the independent models with the same task.

## Q    DISCUSSION ON ADOPTED DATA

The data utilized in this paper are sourced from open-access datasets (*e.g.*, Flowers102 (Nilsback & Zisserman, 2008), SDogs120 (Khosla et al., 2011), ImageNet (Deng et al., 2009)), ptb-text-only (Marcus et al., 1993), and lambada (Paperno et al., 2016). Our research adheres to the terms of their open-source licenses. The ImageNet dataset may include some personal elements, such as human faces. However, our study treats all objects equally and does not intentionally exploit or manipulate these elements. Therefore, our work complies with the requirements of these datasets and should not be construed as a violation of personal privacy.

