# OpenReview forum: "Towards False-claim-resistant Model Ownership Verification via Targeted Fingerprint"
_ICLR.cc/2025/Conference — Submitted to ICLR 2025_

### Official Review · Reviewer_jEMa · 2024-11-01

**Soundness:** 3
**Presentation:** 2
**Contribution:** 3
**Rating:** 6
**Confidence:** 4

**Summary:**

This paper proposes FIT-Print, a defense against false claim attacks in model fingerprinting. The paper first extends false claim attacks to various existing fingerprint schemes and reveals that existing methods are typically vulnerable. The paper attributes this vulnerability to the untargeted nature of existing methods, which allows an adversary to exploit transferrable verification samples to forge a false ownership. The paper then proposes FIT-Print, which features a targeted fingerprint optimization process that aligns the fingerprint to a pre-defined target vector. Based on FIT-Print, the paper provides two concrete implementations, FIT-ModelDiff and FIT-LIME, and experimental results show that the proposed method is effective in verification and robust against false claim attacks.

**Strengths:**

1. **This paper addresses a new and important problem in model fingerprinting**. False claim attack is a relatively new attack in model fingerprinting. This paper provides a defense technique against this new attack.
2. **This paper extends existing false claim attacks to testing-based fingerprint methods**. Previous false claim attacks have mainly focused on adversarial-example-based (AE-based) fingerprints. This paper provides an extension to testing-based methods.
3. **This paper proposes a defense against false claim attacks**. FIT-Print proposes a novel targeted optimization process to craft verification samples w.r.t. a pre-defined target, which limits the transferability of verification samples and is empirically effective against false claims. The method could also be generalized to various testing-based model fingerprints.

**Weaknesses:**

1. **The claim that FIT-Print "restricts the (potential) fingerprint space" lacks theoretical support**. The idea of restricting fingerprint space with targeted optimization is interesting. However, judging from Theorem 1, the false claim attack has little chance of success mainly because the adversary has no knowledge of $F$ and has to make random guesses. It lacks a deeper analysis of whether this design indeed restricts the fingerprint space and how it impacts the optimization of verification samples. Nonetheless, this is understandable considering the complexity of neural networks.
2. **The problem setting of false claim attack is unclear**. The current manuscript lacks a general workflow of the fingerprint registration and verification process. It also lacks a threat model for the false claim attack. Consequently, it is unclear how a fingerprint is registered and verified, or how the adversary could launch a false claim attack. While some of this information could be found in one of the cited works [1], these missing parts would still cause confusion.

[1] Liu et al. "False Claims against Model Ownership Resolution". USENIX Security 2024.

**Questions:**

1. In Section 4.4, is the adversary using the same fingerprint $F$ when creating verification samples from ImageNet independent models?
2. The setting of the adaptive unlearning attack in Section 4.5 is confusing. The authors mention that "the model reuser still has *no* knowledge of the target fingerprint $F$", but the first sentence says "the model reuser *knows* the target fingerprint" and the target fingerprint $F$ is used in the optimization in Eq. 17. It is unclear whether $F$ is known to the adversary or not.
3. What is the overhead of the targeted optimization (fingerprint generation)? The overhead analysis in Appendix F seems to only include the overhead for fingerprint verification. Since the targeted optimization process involves additional optimization and augmented models, will the fingerprint generation process be significantly slower?
4. What is the attack success rate of false claim attacks on existing AE-based or testing-based methods? It seems Table 2 only includes a false positive rate on independent models without considering false claim attacks.
5. From this reviewer's opinion, some discussions in the appendices are quite important, such as the reason for choosing testing-based fingerprints over AE-based fingerprints, the ablation study on the augmented models, and the discussion on the generalization of FIT-Print. Including these in the main part of the paper would help clarify some confusion. This reviewer understands that the authors are restricted by the page limit, but it would be better if some references to the appendices could be put where necessary.

---

> ### Author Response · Authors · 2024-11-19
> **Author Response (Part I)**
>
> Dear Reviewer jEMa, thank you very much for your careful review of our paper and thoughtful comments. We are encouraged by your positive comments on **the importance of the problem we attempt to address and our novelty and contributions**. We hope the following responses can help clarify potential misunderstandings and alleviate your concerns.
>
> ---
>
> **Q1**: The claim that FIT-Print "restricts the (potential) fingerprint space" lacks theoretical support. The idea of restricting fingerprint space with targeted optimization is interesting. However, judging from Theorem 1, the false claim attack has little chance of success mainly because the adversary has no knowledge of $F$ and has to make random guesses. It lacks a deeper analysis of whether this design indeed restricts the fingerprint space and how it impacts the optimization of verification samples. Nonetheless, this is understandable considering the complexity of neural networks.
>
>
> **R1**: Thank you for this insightful comment! We are deeply sorry that our submission may lead you to some misunderstandings that we want to clarify.
>
> - **Theorem 1 is simply used to demonstrate that a random trigger sample is unlikely to falsely claim the ownership**.
> - In particular, **Theorem 1 guides the selection of the verification threshold $\tau$** in our method. Specifically, we can set a proper $\tau$ to make the probability of using a random trigger sample to falsely claim ownership lower than a small security parameter $\kappa$.
> - We admit that our method does not provide proof that demonstrates our method is provably secure. The transferability of our proposed targeted fingerprint is equal to or lower than the untargeted fingerprint but the theoretical gap is not known to us. Empirically, **such a phenomenon is widely validated in the area of adversarial attacks ($i.e.$, targeted adversarial examples have lower transferability than untargeted ones).** The papers from the top peer-reviewed conferences [1,2] also present the proposition. It is also empirically validated as depicted in Table 1 of [3].
>
> We will investigate how to achieve a provably secure model fingerprinting method in our future work.
>
> **Reference**
> 1. Towards Transferable Targeted Attack. CVPR, 2020.
> 2. Towards Transferable Targeted Adversarial Examples. CVPR, 2023.
> 3. LFAA: Crafting Transferable Targeted Adversarial Examples with Low-Frequency Perturbations. arXiv, 2023.

---

> ### Author Response · Authors · 2024-11-19
> **Author Response (Part II)**
>
> **Q2**: The problem setting of false claim attack is unclear. The current manuscript lacks a general workflow of the fingerprint registration and verification process. It also lacks a threat model for the false claim attack. Consequently, it is unclear how a fingerprint is registered and verified, or how the adversary could launch a false claim attack. While some of this information could be found in one of the cited works [1], these missing parts would still cause confusion.
> [1] Liu et al. "False Claims against Model Ownership Resolution". USENIX Security 2024.
>
>
>
> **R2**: Thank you for the insightful comment! We agree that a detailed introduction to the process of model fingerprinting and the threat model of false claim attacks can improve our paper. Accordingly, we add a detailed introduction and a figure in *Appendix A* to clarify those details, as follows.
>
> **A Detailed Threat Models**
>
> In this section, we provide a detailed introduction to the threat models of model fingerprinting and false claim attacks. Three parties involved in the threat models are depicted in Figure 6.
>
> **A.1 Detailed Threat Model of Model Fingerprinting**
>
> There are three parties involved in the threat model of model fingerprinting, including the model developer, the model reuser, and the verifier. The model developer trains a model and the model reuser attempts to steal and reuse this model. The verifier is responsible for fingerprint registration and ownership verification. The assumptions of these three parties can be found in Section 2.1.
>
> **Process of Model Fingerprinting**. Model fingerprinting can be divided into three steps, including fingerprint generation, fingerprint registration, and ownership verification.
>
> 1. **Fingerprint Generation**: In this step, the model developer trains its source model $M_o$ and generates the fingerprint of $M_o$.
> 2. **Fingerprint Registration**: After generating the fingerprint, the model developer registers the fingerprint and the model with a timestamp to a trustworthy third-party verifier.
> 3. **Ownership Verification**: For a suspicious model $M_s$ that could be a reused version of $M_o$, the verifier will first check the timestamps of these two models. If the registration timestamp of $M_s$ is later than $M_o$, the verifier will further check whether the fingerprint of $M_o$ is similar to the fingerprint $M_s$. If so, the suspicious model can be regarded as a reused version of $M_o$.
>
> **A.2 Detailed Threat Model of False Claim Attacks**
>
> There are three parties involved in the threat model of false claim attacks, including the malicious developer, the verifier, and an independent developer. The formal definition of false claim attacks can be found in Section 2.3.
>
> **Assumption of the Malicious Model Developer**. In false claim attacks, the malicious developer is the adversary who aims to craft and register a *transferable* fingerprint to falsely claim the ownership of the independent developer's model $M_I$. The malicious developer is assumed to have adequate computational resources and datasets to train a high-performance model and carefully craft transferable model fingerprints. The primary goal of the malicious developer is that the registered model fingerprints can be verified in as many other models as possible. By generating the transferable fingerprint, the malicious developer can (falsely) claim the ownership of any third-party models (that are registered later than that of the malicious developer).
>
> **Process of False Claim Attacks**. The process of false claim attacks can also be divided into three steps, including fingerprint generation, fingerprint registration, and false ownership verification.
>
> 1. **Fingerprint Generation**: In this step, the model developer trains its source model $M_o$ and attempts to generate a *transferable* fingerprint of $M_o$.
> 2. **Fingerprint Registration**: After generating the fingerprint, the model developer registers the *transferable* fingerprint and the model with a timestamp to a trustworthy third-party verifier.
> 3. **Ownership Verification**: The adversary tries to use the transferable fingerprint to falsely claim the ownership of another independently trained model $M_I$. Since the fingerprint is registered beforehand, the ownership verification won't be rejected due to the timestamp. Subsequently, the benign developer may be accused of infringement.

---

> ### Author Response · Authors · 2024-11-19
> **Author Response (Part III)**
>
> **Q3**: In Section 4.4, is the adversary using the same fingerprint $F$ when creating verification samples from ImageNet independent models?
>
> **R3**: Thank you for the comment! We are deeply sorry that our submission may lead you to some misunderstandings that we want to clarify.
>
> When creating testing samples from independent models, **the adversary utilizes the same fingerprint used for its own model**. This is because **the objective of false claim attacks is to ensure that the fingerprint of the adversary can be verified in other independent models (via transferability)**.
>
>
> We add more details in *Section 4.4* of our revision to avoid potential misunderstandings. Thank you again for pointing it out!
>
> ---
>
> **Q4**: The setting of the adaptive unlearning attack in Section 4.5 is confusing. The authors mention that "the model reuser still has no knowledge of the target fingerprint $F$", but the first sentence says "the model reuser knows the target fingerprint" and the target fingerprint $F$ is used in the optimization in Eq. 17. It is unclear whether $F$ is known to the adversary or not.
>
> **R4**: Thank you for the insightful comment! We are deeply sorry that there appears to be a typo in Section 4.5.
>
> - In the unlearning attack, we assume that **the model reuser knows the target fingerprint of the model developer but has no knowledge of the testing samples**. As such, the model reuser can construct some independent testing samples to unlearn the target fingerprint from the model.
> - We have proofread our manuscript and corrected all typos.
>
> Thank you again for the detailed reading!
>
> ---
>
> **Q5**: What is the overhead of the targeted optimization (fingerprint generation)? The overhead analysis in Appendix F seems to only include the overhead for fingerprint verification. Since the targeted optimization process involves additional optimization and augmented models, will the fingerprint generation process be significantly slower?
>
> **R5**: Thank you for the insightful comment! We hereby provide more detailed discussions to alleviate your concerns.
>
> - In each iteration of optimizing the testing samples, we need to perform one forward propagation and one backward propagation of the fingerprint verification method. Assuming that we utilize $\xi$ augmented models during optimization, **the time complexities of each iteration of testing sample extraction in FIT-ModelDiff and FIT-LIME are $O(\xi\cdot k)$ and $O(\xi\cdot k/\beta)$**. $k$ is the length of the targeted fingerprint and $\beta$ is the batch size.
> - For instance, in our main experiments, we utilize $10$ augmented models and $256$-bit targeted fingerprint. It takes nearly **3 seconds** for one optimization iteration in FIT-ModelDiff and **1 second** for that in FIT-LIME. As such, **our FIT-ModelDiff and FIT-LIME are efficient in optimization and the overhead is acceptable.**
>
> We also add more details in *Appendix H* of our revision.
>
> ---
>
> **Q6**: What is the attack success rate of false claim attacks on existing AE-based or testing-based methods? It seems Table 2 only includes a false positive rate on independent models without considering false claim attacks.
>
> **R6**: Thank you for the insightful comment! We admit that the results of false claim attacks against existing fingerprinting methods are important. We will provide more details to further alleviate your concern.
>
> - For testing-based methods, we design a false claim attack in Section 2.3. The experimental results are shown in Table 1 below, demonstrating that **performing false claim attacks can significantly increase the false positive rates**.
> - For AE-based methods, designing a false claim attack is equivalent to designing a transferable adversarial attack. We utilize the false claim attack proposed in [1] and **the results in Table 1 show that the false claim attack is also effective to AE-based methods**.
>
> We also add more details in *Appendix E* of our revision.
>
>
> **Table 1**: The results of false claim attack against existing model fingerprinting methods.
>
>
> | Method$\rightarrow$ | IPGuard | ModelDiff | Zest | SAC | FIT-ModelDiff (ours) | FIT-LIME (ours) |
> | -------- | -------- | -------- | ----------|---------| ----------|---------|
> | False Positive Rate     | 30.6% | 4.0%     | 7.6%   |  39.6% | 0.0% | 0.0% |
> | False Positive Rate After Attacks| 61.8% | 15.28% | 29.51% | 51.94% | 0.0% | 0.0% |
>
> **Reference**
> 1. False Claims against Model Ownership Resolution. USENIX Security, 2024.

---

> ### Author Response · Authors · 2024-11-19
> **Author Response (Part IV)**
>
> **Q7**: From this reviewer's opinion, some discussions in the appendices are quite important, such as the reason for choosing testing-based fingerprints over AE-based fingerprints, the ablation study on the augmented models, and the discussion on the generalization of FIT-Print. Including these in the main part of the paper would help clarify some confusion. This reviewer understands that the authors are restricted by the page limit, but it would be better if some references to the appendices could be put where necessary.
>
> **R7**: Thank you for the insightful comment! We agree that some experiments and discussions are important in our work but we have to put it into the appendix due to the page limit. We also agree that adding references to the appendices in the main part of the manuscript is necessary. As such, we make the following changes in our revision to highlight them:
>
> - We have summarized all the experiments and discussions, including ablation studies and discussion on the generalization, at the beginning of the experiments section.
> - We add references to experimental details in Section 4.1.
> - We add references to discussions on more related works in Section 2.2.
>
> If we miss something important and do not include the right reference to them in our main part, please let us know. We are very willing to add such references :)

---

> ### Author Response · Authors · 2024-11-23
> **Thanks to Reviewer jEMa**
>
> Please allow us to thank you again for reviewing our paper and the valuable feedback, and in particular for recognizing the strengths of our paper in terms of the importance of the problem we attempt to address and our novelty and contributions.
>
> Please let us know if our response and the new experiments have properly addressed your concerns. We are more than happy to answer any additional questions during the discussion period. Your feedback will be greatly appreciated.

---

> > ### Comment · Reviewer_jEMa · 2024-11-24
> >
> > This reviewer would like to thank the author(s) for their detailed and timely response. The response has addressed most of the concerns. The reviewer would like to change the score to 6 (marginally above acceptance threshold), based on the following considerations:
> >
> > 1. The paper addresses a timely and important topic of defending against false claim attacks in model fingerprinting.
> > 2. The problem setting of false claim attack (along with a few other minor issues) have been clarified in the revised manuscript. A detailed workflow of model fingerprinting and false claim attack has been added to Appendix A, which resolves one of this reviewer's major concerns.
> > 3. Still, the current manuscript lacks a deeper analysis or evaluation on the transferability of the targeted fingerprints. This reviewer agrees that empirically, targeted adversarial examples tend to have lower transferability. However, it is unclear whether a more sophisticated adversary could forge a false claim on FIT-Print (e.g., by exploiting techniques from transferrable targeted adversarial attacks). Apart from a lack of formal proof (which is a noteworthy but understandable limitation, given the complexity of neural networks), the current adaptive evaluation is also limited (in Section 4.4, the adaptive attacker only leverages additional independent models). Hence, a more in-depth analysis or evaluation on this aspect would help make the claim more persuasive.

---

> > > ### Author Response · Authors · 2024-11-24
> > > **Thank You for Your Positive Feedback!**
> > >
> > > Dear Reviewer jEMa:
> > >
> > > Thank you so much for your positive feedback! It encourages us a lot.
> > >
> > > We agree that our approach is only a significant step toward addressing false claim attacks rather than a thorough solution to the problem. For instance, we fail to provide a formal guarantee of its resistance against false claim attacks, although we have empirically verified it. As such, a more powerful adversary may still be able to conduct successful false claim attacks. We add an additional discussion on it in *Appendix  O*, as follows.
> > >
> > > ```!
> > > Another potential limitation is that FIT-Print does not provide formal proof of the resistance to false claim attacks. As such, a more powerful adversary may still be able to conduct a successful false claim attack. We will investigate how to achieve a certified robust model fingerprinting method against false claim attacks in our future work.
> > > ```
> > >
> > > We hope that our work can be a concrete step towards false-claim-resistant model fingerprinting and inspire subsequent works.
> > >
> > > Thank you again for your valuable time and insightful review!
> > >
> > > Best regards,
> > >
> > > Paper 5867 Author(s)

---

### Official Review · Reviewer_LMaC · 2024-11-01

**Soundness:** 2
**Presentation:** 1
**Contribution:** 1
**Rating:** 1
**Confidence:** 3

**Summary:**

To protect an AI model's copyright from illegal reuse or plagiarism, a method to verify true ownership is required. A fingerprinting strategy can be used without altering the model, preserving its performance. However, adversarial modifications used for fingerprinting can enable false claim attacks, where an attacker falsely claims ownership by exploiting the transferability of these modifications.
To counter false claim attacks, the authors propose a targeted adversarial fingerprinting method. They argue that current techniques are vulnerable due to their untargeted nature, which creates a large fingerprint space, enabling adversarial transfers between models. By focusing on specific target classes, they suggest that the fingerprint space can be narrowed, reducing transferability and minimizing false claim risks.

**Strengths:**

Supplementary Materials
The supplementary materials include code and a clear README, which demonstrates a responsible effort towards transparency and reproducibility.

**Weaknesses:**

1. Contribution and Innovation Deficiency
Contribution:
There are some issues with the contributions mentioned in the paper:
Contribution 1: False claim attack is not a new type of attack. This type of attack has already been introduced in previous work, and thus, it should not be considered a contribution of this paper.
Contribution 2: The advantages of the proposed targeted verification method (compared to other verification methods) have not been sufficiently explained. Without clearly demonstrating the superiority of the targeted approach over other methods, it is difficult to view this as a valid contribution.
Contribution 3: The advantages of the proposed black-box fingerprinting method (compared to existing methods) are also insufficiently explained. Without providing more explanation to highlight the uniqueness or strengths of this method, it cannot be considered a significant contribution.
Novelty:
The main innovation of the paper seems to be the shift from non-targeted to targeted fingerprinting methods. However, this innovation appears somewhat limited and lacks sufficient persuasive power. Beyond this point, the paper's contributions in terms of innovation could be further enhanced. It is recommended to explore additional novel ideas to strengthen the overall impact of the paper.

2. Terminology and Definitions are Unclear
Improper naming:
The choice of abbreviations in the paper is not standard, especially in "False-claIm-resistant Targeted model fingerPrinting (FIT-Print)." Using the letter "I" from the middle of the word "claim" is unconventional and does not follow common abbreviation rules. Naming should avoid such arbitrary choices to ensure clarity and consistency, and it is recommended to use abbreviations that are more in line with standard conventions.
Unclear explanation of the verification process:
In section 2.3, the explanation of ownership verification through "p1" and "p2" is overly brief. Using such abbreviations makes it difficult for readers to grasp their exact meaning. It is recommended to clearly explain each step of the verification process and avoid using vague abbreviations that could lead to confusion.
Unclear definition of independent model:
The term "independent model" is used in the paper but not clearly defined. Does "independent" refer to models with completely different structures, or models performing different tasks? This distinction is critical for understanding the relationships between models in the verification scenario. It is recommended to clearly define whether independence is based on structure, task, or dataset.
Unclear definition of the scenario and protocol:
The paper does not clearly define each party involved in the fingerprint verification scenario, nor the specific steps they execute in the protocol. For instance, the roles of each party, the information they have access to, and the actions they perform should be clearly explained, so that readers can better understand how the entire verification protocol works. Similarly, providing more specific steps and details for both the attack and defense mechanisms—such as the inputs, outputs, who executes each step—would greatly improve the understanding of these processes.
Unclear model relationships:
The symbols for models (e.g., Mo, Ms, Mi) are not clearly explained, particularly in the context of the "false claim of ownership of M1." Providing clearer descriptions and diagrams would improve understanding of how these models interact in the scenario.
Lack of illustrations:
The paper lacks sufficient diagrams to explain the overall fingerprint verification framework. The relationships between models like Mo, Ms, and Mi are unclear, making it harder for readers to follow. Diagrams would help visualize these interactions and clarify the verification process.

Be careful when use of terminology:
Terms like "Proposition," "Definition," and "Theorem" are used in the paper but need to be applied with more caution. It is recommended to briefly introduce these terms in the introduction and clarify their specific contexts throughout the paper to help readers better understand their relevance and contribution to the argument.

3. Concerns about the Mechanism or Inadequate Consideration/Explanation of Vulnerabilities.
Fingerprinting scenario considerations:
The paper does not fully explore whether adversarial samples generated by the model owner could also affect models not owned by them, given the known transferability of such samples. This scenario needs further investigation to assess its potential impact on the verification process.
Fairness in adversarial sample generation:
The process for generating adversarial samples x̄ and x is unclear. If different algorithms are used, it raises concerns about fairness in the verification process. Ideally, fingerprinting rules should be enforced by a trusted third party to ensure consistency. Clarifying the generation algorithms would enhance transparency and fairness.

**Questions:**

The weaknesses of the paper have been pointed out in the Weaknesses section, and I hope that targeted revisions can be made for each issue. Overall, I believe the following improvements would enhance the quality of the paper:

1. Clear explanation and presentation
Please provide detailed explanations of the roles, tasks, and relationships of each party in the scenario, along with a step-by-step description of the process. I also expect more elaboration on potential attacks and defenses, specifying the processes involved.
I encourage the authors to invest more effort in improving the readability of the paper, such as by incorporating diagrams to help readers understand the proposed protocol and background knowledge. The terms and concepts used should be clearly defined, as omitting details here could compromise the clarity of the paper. Please also avoid using non-standard abbreviations or expressions that may confuse readers.

2. Accurate and expanded contributions
Accurately and truthfully present the contributions of your work. If the current contributions are insufficient, consider adding new ones. I believe there is room for further enhancement in the paper's contributions to innovation.

3. Explanation and discussion
This point corresponds to my concerns about the mechanism or inadequate consideration/explanation of vulnerabilities. I hope the authors can alleviate these concerns by providing additional explanations, analyses, or clear descriptions of the experimental setup. This would enhance the reliability of the work presented in the paper.

---

> ### Author Response · Authors · 2024-11-19
> **Author Response (Part I)**
>
> Dear Reviewer LMaC, thank you very much for your review of our paper. We are encouraged by your positive comments on our **efforts regarding transparency and reproducibility**. We hope the following responses can help clarify potential misunderstandings and alleviate your concerns.
>
> ---
>
> **Q1-1**: Contribution and Innovation Deficiency Contribution: There are some issues with the contributions mentioned in the paper: Contribution 1: False claim attack is not a new type of attack. This type of attack has already been introduced in previous work, and thus, it should not be considered a contribution of this paper.
>
> **R1-1**: Thank you for the comment. However, we believe there is some potential misunderstanding.
> - In our paper, **we do not claim that we propose a new type of attack**. Instead, as clearly stated in our contributions (Line 89-90), we revisit existing model fingerprinting methods and **design an effective attack implementation** to reveal that existing methods are vulnerable to false claim attacks.
> - We also clearly stated that 'the concept and definition of false claim attacks were initially introduced in [1] and primarily targeted at attacking model watermarking methods' in our first footnote (Line 107).
>
> If you can kindly provide the related reference, we are also willing to make further clarification.
>
> ---
>
> **Q1-2**: Contribution 2: The advantages of the proposed targeted verification method (compared to other verification methods) have not been sufficiently explained. Without clearly demonstrating the superiority of the targeted approach over other methods, it is difficult to view this as a valid contribution.
>
> **R1-2**: Thank you for the comment.
>
> - **Our methods outperform existing methods primarily due to the proposed FIT-Print paradigm.** Our FIT-Print leverages *targeted* fingerprinting to decrease the transferability of fingerprints and resist false claim attacks. Our main insight is that untargeted fingerprints enlarge the space of viable fingerprints while **targeted fingerprints can decrease the transferability and reduce the probability of successful false claim attacks**.
> - We empirically validate the advantages of our proposed methods in *Section 4*.
>
> If you can kindly provide more detailed information about which aspect we do not sufficiently explain, we are also willing to make further clarification.
>
>
> ---
>
> **Q1-3**: Contribution 3: The advantages of the proposed black-box fingerprinting method (compared to existing methods) are also insufficiently explained. Without providing more explanation to highlight the uniqueness or strengths of this method, it cannot be considered a significant contribution.
>
> **R1-3**: Thank you for the comment. **Our proposed black-box fingerprinting methods, FIT-ModelDiff and FIT-LIME, outperform existing methods primarily due to the proposed FIT-Print paradigm**. They are two concrete implementations under FIT-Print. FIT-ModelDiff and FIT-LIME are representatives of the bit-wise method (extract the fingerprint bit by bit) and list-wise method (extract the fingerprint as a whole). We empirically evaluate the effectiveness of FIT-ModelDiff and FIT-LIME in *Section 4*.
>
> If you can kindly provide more detailed information about the potential weaknesses of our proposed methods, we are also willing to make further clarification.

---

> ### Author Response · Authors · 2024-11-19
> **Author Response (Part II)**
>
> **Q1-4**: Novelty: The main innovation of the paper seems to be the shift from non-targeted to targeted fingerprinting methods. However, this innovation appears somewhat limited and lacks sufficient persuasive power. Beyond this point, the paper's contributions in terms of innovation could be further enhanced. It is recommended to explore additional novel ideas to strengthen the overall impact of the paper.
>
> **R1-4**: Thank you for your comment. We respectfully disagree with this argument.
>
> - Our contributions are **not limited to the shift from non-targeted to targeted** fingerprinting methods.
>     - We **revisit existing model fingerprinting methods and design an implementation of false claim attacks**. We reveal that existing methods are vulnerable to false claim attacks.
>     - We **introduce a new fingerprinting paradigm**, where we conduct fingerprint verification in a targeted manner. Our main insight is that untargeted fingerprints enlarge the space of viable fingerprints while targeted fingerprints can decrease the transferability and reduce the probability of successful false claim attacks.
>     - Based on our proposed paradigm, we **design two advanced black-box targeted model fingerprinting methods**, FIT-ModelDiff and FIT-LIME. FIT-ModelDiff and FIT-LIME are representatives of the bit-wise method (extract the fingerprint bit by bit) and list-wise method (extract the fingerprint as a whole), respectively.
>     - We conduct experiments on 2 benchmark datasets, 2 models, and 6 baseline fingerprinting methods to verify the effectiveness and conferability of our FIT-Print and its resistance to false claims and adaptive attacks.
> - We admit that our methods appear to be simple, but **it does not mean that our methods do not have sufficient novelties and contributions**.
>     - **Our method is simple yet effective since our insight is deep and thorough**.
>     - **We believe that simplicity is not necessarily a bad thing** because the follow-up researchers and developers can easily implement our methods and paradigm to further promote model copyright protection.
>
>
> If you could provide more detailed information about your concerns, we are also willing to make further clarification.
>
> ---
>
> **Q2**: Terminology and Definitions are Unclear Improper naming: The choice of abbreviations in the paper is not standard, especially in "False-claIm-resistant Targeted model fingerPrinting (FIT-Print)." Using the letter "I" from the middle of the word "claim" is unconventional and does not follow common abbreviation rules. Naming should avoid such arbitrary choices to ensure clarity and consistency, and it is recommended to use abbreviations that are more in line with standard conventions.
>
> **R2**: Thank you for the comment. Our proposed paradigm is named FIT-Print. 'FIT' may correspond to the core of our paradigm, $i.e.,$ fitting to a targeted fingerprint. Besides, we do not agree that naming can only utilize the first letter.
>
> If you can kindly provide authoritative documents that the name of a method must use the first letter, we are willing to make a change.
>
>
> ---
>
> **Q3**: Unclear explanation of the verification process: In section 2.3, the explanation of ownership verification through "p1" and "p2" is overly brief. Using such abbreviations makes it difficult for readers to grasp their exact meaning. It is recommended to clearly explain each step of the verification process and avoid using vague abbreviations that could lead to confusion.
>
> **R3**: Thank you for the comment. However, we do not use 'p1' or 'p2' in Section 2.3. Instead, we use the full names 'Proposition 1' and 'Proposition 2'. We believe they are referenced explicitly.
>
> If you can kindly provide more information regarding 'p1' and 'p2', we are willing to make further clarification.

---

> ### Author Response · Authors · 2024-11-19
> **Author Response (Part III)**
>
> **Q4**: Unclear definition of independent model: The term "independent model" is used in the paper but not clearly defined. Does "independent" refer to models with completely different structures, or models performing different tasks? This distinction is critical for understanding the relationships between models in the verification scenario. It is recommended to clearly define whether independence is based on structure, task, or dataset.
>
> **R4**: Thank you for the insightful comment! We are deeply sorry that our submission may lead you to some misunderstandings that we want to clarify.
>
> - **The independent models are defined as the models that are independently trained by other parties.** The independent models may have different model architectures or different datasets from the source model.
> - We note that following prior works [1,2,3], **fine-tuned models, pruned models, transfer learning models, and extracted models are regarded as reused models**.
>
> **Reference**
> 1. Ipguard: Protecting Intellectual Property of Deep Neural Networks via Fingerprinting the Classification Boundary. AsiaCCS, 2021.
> 2. ModelDiff: Testing-based DNN Similarity Comparison for Model Reuse Detection. SIGSOFT, 2021.
> 3. Are You Stealing My Model? Sample Correlation for Fingerprinting Deep Neural Networks. NeurIPS, 2022.

---

> ### Author Response · Authors · 2024-11-19
> **Author Response (Part IV)**
>
> **Q5**: Unclear definition of the scenario and protocol: The paper does not clearly define each party involved in the fingerprint verification scenario, nor the specific steps they execute in the protocol. For instance, the roles of each party, the information they have access to, and the actions they perform should be clearly explained, so that readers can better understand how the entire verification protocol works. Similarly, providing more specific steps and details for both the attack and defense mechanisms—such as the inputs, outputs, who executes each step—would greatly improve the understanding of these processes.
>
>
> **R5**: Thank you for the comment. We agree that a detailed introduction to the process of model fingerprinting and the threat model of false claim attacks can improve our paper.
>
> - **We have discussed the assumptions and roles of different parties in *Section 2.1***.
> - **We also add a detailed introduction and a figure in *Appendix A* to clarify the threat models and processes**, as follows.
>
> **A Detailed Threat Models**
>
> In this section, we provide a detailed introduction to the threat models of model fingerprinting and false claim attacks. Three parties involved in the threat models are depicted in Figure 6.
>
> **A.1 Detailed Threat Model of Model Fingerprinting**
>
> There are three parties involved in the threat model of model fingerprinting, including the model developer, the model reuser, and the verifier. The model developer trains a model and the model reuser attempts to steal and reuse this model. The verifier is responsible for fingerprint registration and ownership verification. The assumptions of these three parties can be found in Section 2.1.
>
> **Process of Model Fingerprinting**. Model fingerprinting can be divided into three steps, including fingerprint generation, fingerprint registration, and ownership verification.
>
> 1. **Fingerprint Generation**: In this step, the model developer trains its source model $M_o$ and generates the fingerprint of $M_o$.
> 2. **Fingerprint Registration**: After generating the fingerprint, the model developer registers the fingerprint and the model with a timestamp to a trustworthy third-party verifier.
> 3. **Ownership Verification**: For a suspicious model $M_s$ that could be a reused version of $M_o$, the verifier will first check the timestamps of these two models. If the registration timestamp of $M_s$ is later than $M_o$, the verifier will further check whether the fingerprint of $M_o$ is similar to the fingerprint $M_s$. If so, the suspicious model can be regarded as a reused version of $M_o$.
>
> **A.2 Detailed Threat Model of False Claim Attacks**
>
> There are three parties involved in the threat model of false claim attacks, including the malicious developer, the verifier, and an independent developer. The formal definition of false claim attacks can be found in Section 2.3.
>
> **Assumption of the Malicious Model Developer**. In false claim attacks, the malicious developer is the adversary who aims to craft and register a *transferable* fingerprint to falsely claim the ownership of the independent developer's model $M_I$. The malicious developer is assumed to have adequate computational resources and datasets to train a high-performance model and carefully craft transferable model fingerprints. The primary goal of the malicious developer is that the registered model fingerprints can be verified in as many other models as possible. By generating the transferable fingerprint, the malicious developer can (falsely) claim the ownership of any third-party models (that are registered later than that of the malicious developer).
>
> **Process of False Claim Attacks**. The process of false claim attacks can also be divided into three steps, including fingerprint generation, fingerprint registration, and false ownership verification.
>
> 1. **Fingerprint Generation**: In this step, the model developer trains its source model $M_o$ and attempts to generate a *transferable* fingerprint of $M_o$.
> 2. **Fingerprint Registration**: After generating the fingerprint, the model developer registers the *transferable* fingerprint and the model with a timestamp to a trustworthy third-party verifier.
> 3. **Ownership Verification**: The adversary tries to use the transferable fingerprint to falsely claim the ownership of another independently trained model $M_I$. Since the fingerprint is registered beforehand, the ownership verification won't be rejected due to the timestamp. Subsequently, the benign developer may be accused of infringement.

---

> ### Author Response · Authors · 2024-11-19
> **Author Response (Part V)**
>
> **Q6**: Unclear model relationships: The symbols for models (e.g., Mo, Ms, Mi) are not clearly explained, particularly in the context of the "false claim of ownership of M1." Providing clearer descriptions and diagrams would improve understanding of how these models interact in the scenario.
>
> **R6**: Thank you for the comment! Hereby we provide further clarification about the symbols for models.
>
> - $M_o$: the original (or source) model which is developed by the model owner. $M_o$ is the model we need to protect.
> - $M_s$: the suspicious model which is suspected to be reused from the source model $M_o$.
> - $M_i$: the independently trained model owned by other parties.
>
> The subscripts of these symbols are the first letters of 'original', 'suspicious', and 'independent'. We have explained the definitions of these symbols in *Section 2*.
>
> ---
>
>
> **Q7**: Lack of illustrations: The paper lacks sufficient diagrams to explain the overall fingerprint verification framework. The relationships between models like Mo, Ms, and Mi are unclear, making it harder for readers to follow. Diagrams would help visualize these interactions and clarify the verification process.
>
>
> **R7**: Thank you for the comment. We have integrated the introduction to the relationships between $M_o, M_s, M_I$ in Figure 6 in Appendix A. We also provide detailed clarification of the process of fingerprint verification. Please kindly refer to **R5&6** for more details.
>
>
> ---
>
> **Q8**: Be careful when use of terminology: Terms like "Proposition," "Definition," and "Theorem" are used in the paper but need to be applied with more caution. It is recommended to briefly introduce these terms in the introduction and clarify their specific contexts throughout the paper to help readers better understand their relevance and contribution to the argument.
>
> **R8**: Thank you for the comment. We respectfully note that we have strictly followed the definitions and rules in mathematics [1] to utilize the terms 'Proposition', 'Definition', and 'Theorem'. We believe these terms are common sense and overinterpretation of these terms may weaken the focus of our paper.
>
> **Reference**
> 1. Mathematical Writing. 1989.
>
> ---
>
> **Q9**: Concerns about the Mechanism or Inadequate Consideration/Explanation of Vulnerabilities. Fingerprinting scenario considerations: The paper does not fully explore whether adversarial samples generated by the model owner could also affect models not owned by them, given the known transferability of such samples. This scenario needs further investigation to assess its potential impact on the verification process.
>
> **R9**: Thank you for the comment. We hereby make further clarification to alleviate your concerns.
>
> - **The issue of transferability is actually the core of our study.** False claim attacks aim to craft transferable fingerprints so that the adversary can falsely claim the ownership of other parties' models. The success of false claim attacks depends on the transferability of model fingerprints.
> - **We tackle the issue of false claim attacks ($i.e.$, transferability) via targeted fingerprinting.** Our main insight is that untargeted fingerprints enlarge the space of viable fingerprints while targeted fingerprints can decrease the transferability and reduce the probability of successful false claim attacks. Arguably, **such an insight can also be found in the area of adversarial attacks ($i.e.$, targeted adversarial examples have lower transferability than untargeted ones).** The papers from the top peer-reviewed conferences [1,2] also present the proposition. It is also empirically validated as depicted in Table 1 of [3].
> - **We empirically validate that our proposed methods have low false positive rates and can resist adaptive false claim attacks**. The experimental results are shown in *Section 4.2&4.4*.
>
> **Reference**
> 1. Towards Transferable Targeted Attack. CVPR, 2020.
> 2. Towards Transferable Targeted Adversarial Examples. CVPR, 2023.
> 3. LFAA: Crafting Transferable Targeted Adversarial Examples with Low-Frequency Perturbations. arXiv, 2023.

---

> ### Author Response · Authors · 2024-11-19
> **Author Response (Part VI)**
>
> **Q10**: Fairness in adversarial sample generation: The process for generating adversarial samples x̄ and x is unclear. If different algorithms are used, it raises concerns about fairness in the verification process. Ideally, fingerprinting rules should be enforced by a trusted third party to ensure consistency. Clarifying the generation algorithms would enhance transparency and fairness.
>
> **R10**: Thank you for the comment. We will make further clarification to alleviate your concern.
>
> We argue that **the comparison between different fingerprinting methods in our paper is fair.**
> - We **use the open-source code and the default settings provided in the papers for the experiments**. We also utilize a unified benchmark of model reuse.
> - **The primary advantage of our method is due to the paradigm of using targeted fingerprints instead of simply a new generation algorithm**. Targeted fingerprints can decrease the transferability and reduce the probability of successful false claim attacks. The advantage is regardless of the settings ($e.g.$, optimization methods).

---

> ### Author Response · Authors · 2024-11-23
> **Thanks to Reviewer LMaC**
>
> Please allow us to thank you again for reviewing our paper and the feedback, and in particular for recognizing the strengths of our paper in terms of our efforts regarding transparency and reproducibility.
>
> Please let us know if our response and the new experiments have properly addressed your concerns. We are more than happy to answer any additional questions during the discussion period. Your feedback will be greatly appreciated.

---

### Official Review · Reviewer_oTwh · 2024-11-03

**Soundness:** 2
**Presentation:** 3
**Contribution:** 2
**Rating:** 3
**Confidence:** 5

**Summary:**

This paper studied the vulnerability of existing model fingerprinting approaches against false claim attacks. To fix the problem, they propose to incorporate a mapping function, which can be confidential, and input perturbation, which caters to the mapping, to defend against such attacks.

**Strengths:**

- The paper is well-written and has plenty of experiments to support the contributions.
- The research question is interesting.

**Weaknesses:**

- The attack scenario is limited. Currently, to the best of my knowledge, there is no third-party management on the fingerprints to authorize the ownership of open-sourced models. Therefore, it is impractical to consider and even defend against false-claim attacks. The authors should elaborate on the practical side of the setting.
- Another major concern is the missing literature. There is a work on KDD23 which also leverages a mapping function (they call it a meta-verifier) and adaptive input samples to build the model fingerprint. What is the different in methodology? The authors did not even mention the work. They should at least compare with it and show the substantial contribution over the work.

[1] https://dl.acm.org/doi/10.1145/3534678.3539257

**Questions:**

See the weakness part above.

---

> ### Author Response · Authors · 2024-11-19
> **Author Response (Part I)**
>
> Dear Reviewer oTwh, thank you very much for your careful review of our paper and thoughtful comments. We are encouraged by your positive comments on **our clear writing**, **the interesting research question**, and **the thorough experiments**. We hope the following responses can help clarify potential misunderstandings and alleviate your concerns.
>
> ---
>
> **Q1**: The attack scenario is limited. Currently, to the best of my knowledge, there is no third-party management on the fingerprints to authorize the ownership of open-sourced models. Therefore, it is impractical to consider and even defend against false-claim attacks. The authors should elaborate on the practical side of the setting.
>
> **R1**: Thank you for the insightful comment! We will clarify the potential misunderstanding to alleviate your concern.
>
> - **The false-claim attack exists even without third-party fingerprint management**.
> - Arguably, **introducing a third party to manage the model copyrights and the fingerprints is rational and necessary in practice.**
>     - **Establishing the artificial intelligence regulator (AIR) is a political trend.** Many countries and regions are in the process of establishing or have established AIR to regulate AI models in various aspects including **copyright**, safety, and transparency ($e.g.$, as exemplified in the EU Artificial Intelligence Act). We present a detailed discussion in *Appendix M*.
>     - **Not establishing a third-party regulator can make false claim attacks much easier.** This proposition is also presented in [1, 2]. Registering the model and its fingerprint to the third-party regulator with a timestamp can prevent any false claim after registration. During ownership verification, the forged fingerprint with a later timestamp will be regarded as invalid. As such, false claim attacks hinge on generating and registering a transferable fingerprint beforehand. This is actually what we attempt to address in this paper.
> - Besides improving the resistance of model fingerprinting against false claim attacks, **our FIT-Print also achieves the SOTA effectiveness regarding detecting reused models**. This is empirically validated in our experiments as shown in *Table 2* in *Section 4.2*.
> - Although currently there is no third-party management on the fingerprints, we believe that our work is still meaningful due to plausible scenarios and foreseeable future needs. Arguably, research is supposed to be ahead of applications.
>
> **Reference**
> 1. False Claims against Model Ownership Resolution. USENIX Security, 2024.
> 2. GrOVe: Ownership Verification of Graph Neural Networks using Embeddings. S&P, 2024.

---

> ### Author Response · Authors · 2024-11-19
> **Author Response (Part II)**
>
> **Q2**: Another major concern is the missing literature. There is a work on KDD23 which also leverages a mapping function (they call it a meta-verifier) and adaptive input samples to build the model fingerprint. What is the different in methodology? The authors did not even mention the work. They should at least compare with it and show the substantial contribution over the work.
> [1] https://dl.acm.org/doi/10.1145/3534678.3539257
>
>
> **R2**: Thank you for suggesting the outstanding work! We are deeply sorry for missing this paper. We hereby make further clarification to alleviate your concerns.
>
> - **Brief Introduction to MetaV**: MetaV[1] introduces two critical components, the adaptive fingerprint and the meta-verifier. The adaptive fingerprint is a set of adversarial examples. The meta-verifier takes the suspicious model's output of the adaptive fingerprint and outputs whether the suspicious model is reused from the original model. MetaV accomplishes such an objective by simultaneously optimizing the adaptive fingerprint ($i.e.$, adversarial perturbations) and the meta-verifier ($i.e.$, a fully-connected neural network). In conclusion, MetaV provided a task-agnostic fingerprinting framework. **MetaV can be regarded as one of the adversarial example-based (AE-based) fingerprinting methods**.
> - **The Advantages of our FIT-Print over MetaV**:
>     - **MetaV is vulnerable to false claim attacks.** MetaV is an AE-based fingerprinting method and the adversary can craft transferable adversarial examples to achieve false claim attacks. This proposition is also presented in [2].
>     - **MetaV cannot detect transfer learning models**, which is one of the realistic stealing settings. MetaV depends on a pre-trained meta-verifier. Transfer learning models may have different output formats, $e.g.$, the number of classes. Therefore, the meta-verifier which has a fixed input format is not able to process the changed outputs of the suspicious model and detect whether it is reused from the original model.
> - We also empirically evaluate the effectiveness of MetaV. Table 1 shows that **our FIT-ModelDiff and FIT-LIME outperform all the baseline methods (including MetaV)**.
>
>
> We sincerely thank you again for the provided outstanding reference. We also add a detailed comparison to this work in *Appendix M.2* of our revision. If you have any suggestions on other references, we are also willing to make further clarifications and discussions :)
>
>
>
>
> **Table 1.** Successful ownership verification rates of different model fingerprinting methods.
>
> | Reuse Task$\downarrow$ | #Models$\downarrow$ | IPGuard | MetaV | ModelDiff | Zest | SAC | ModelGiF | FIT-ModelDiff (ours) | FIT-LIME (ours) |
> | -------- | -------- | -------- |-------- | -------- | -------- | -------- |-------- |-------- |-------- |
> | Copying     | 4     | 100%     | 100%     | 100%     | 100%     | 100%     | 100%   | 100%     | 100%   |
> | Fine-tuning     | 12     | 100%     | 100%     | 100%     | 100%     | 100%     | 100%   | 100%     | 100%   |
> | Pruning     | 12     | 100%     | 100%     | 100%     | 91.67%     | 100%     | 100%   | 100%     | 100%   |
> | Extraction     | 8     | *50%*     | 87.5%     | *50%*     | *25%*     | 100%     | 100%   | 100%     | 100%   |
> | Transfer     | 12     | *N/A*     | *N/A*     | 100%     | *N/A*     | *0%*     | 100%   | 100%     | 100%   |
> | Independent     | 144     | 30.6%     | 4.8%     | 4.0%     | 7.6%     | 39.6%     | 0.0%   | 0.0%     | 0.0%   |
>
>
> **Reference**
> 1. MetaV: A Meta-Verifier Approach to Task-Agnostic Model Fingerprinting. SIGKDD, 2022.
> 2. False Claims against Model Ownership Resolution. USENIX Security, 2024.

---

> ### Author Response · Authors · 2024-11-23
> **Thanks to Reviewer oTwh**
>
> Please allow us to thank you again for reviewing our paper and the valuable feedback, and in particular for recognizing the strengths of our paper in terms of our clear writing, the interesting research question, and the thorough experiments.
>
> Please let us know if our response and the new experiments have properly addressed your concerns. We are more than happy to answer any additional questions during the discussion period. Your feedback will be greatly appreciated.

---

> > ### Author Response · Authors · 2024-11-25
> > **A Gentle Reminder of the Post-rebuttal Feedback**
> >
> > Thank you very much again for your initial comments. They are very valuable for improving our work. We would be grateful if you could have a look at our response and modifications and please let us know if there is anything else that can be added to our next version. We are also willing to have further discussions with you.

---

> > > ### Author Response · Authors · 2024-11-28
> > > **A Second Reminder of the Post-rebuttal Feedback**
> > >
> > > Dear Reviewer oTwh,
> > >
> > > We greatly appreciate your initial comments. We totally understand that you may be extremely busy at this time. But we still hope that you could have a quick look at our responses to your concerns. We appreciate any feedback you could give us. We also hope that you could kindly update the rating if your questions have been addressed. We are also happy to answer any additional questions before the rebuttal ends.
> > >
> > > Best Regards,
> > >
> > > Paper 5867 Author(s)

---

> > > > ### Author Response · Authors · 2024-12-02
> > > > **Reminder of the Post-rebuttal Feedback and Summary of Our Response**
> > > >
> > > > Dear Reviewer oTwh,
> > > >
> > > > Thank you for your time and effort in evaluating our work. We greatly appreciate your initial comments. Your insights and suggestions are extremely valuable to us.
> > > >
> > > > Given that we have only *one day* left for discussion, we are hoping to receive any additional feedback or question you might have at your earliest convenience. We totally understand that you may be busy at this time. But we still hope that you could have a quick look at our responses to your concerns. Your expertise would be of great help to us in improving the quality and rigor of our work.
> > > >
> > > > To facilitate the discussion, we would like to summarize our response as follows.
> > > >
> > > > - **We clarified the rationality and necessity of our setting and threat model**. Introducing a third party to manage the model copyrights is necessary in practice and the false claim attack investigated in our paper also exists even without the third party. Our work is meaningful due to plausible scenarios and foreseeable future needs.
> > > > - **We discussed the suggested related work MetaV [1] and empirically compared it with our method**. The experimental results demonstrated that our method outperforms MetaV.
> > > >
> > > > If our responses address your concerns, we kindly request that you reconsider your evaluations. We would also be grateful for any additional comments or suggestions you might have to refine our work.
> > > >
> > > > Best regards,
> > > >
> > > > Paper 5867 Author(s)
> > > >
> > > > **Reference**
> > > >
> > > > 1. MetaV: A Meta-Verifier Approach to Task-Agnostic Model Fingerprinting. SIGKDD, 2022.

---

### Official Review · Reviewer_toL1 · 2024-11-03

**Soundness:** 3
**Presentation:** 2
**Contribution:** 3
**Rating:** 6
**Confidence:** 3

**Summary:**

This paper addresses the challenge of protecting intellectual property rights (IPR) in open-source pre-trained models, which are widely reused but often without authorization. The authors pointed out a vulnerability in existing fingerprinting techniques, i.e., they are prone to false claim attacks, where adversaries could falsely claim ownership of third-party models. To address this, the authors propose a targeted fingerprinting paradigm called FIT-Print, designed specifically to counteract false claim attacks. FIT-Print transforms model fingerprints into targeted signatures through optimization. Based on this paradigm, the authors develop two black-box fingerprinting methods—FIT-ModelDiff and FIT-LIME, which rely on bit-wise and list-wise approaches, respectively, by leveraging the output distances and feature attributions of specific samples as fingerprints. Experiments on benchmark models and datasets demonstrate the effectiveness of the FIT-Print approach, showing that it is robust to false claim attacks.

**Strengths:**

+) Defending against false claim attacks is a timely topic.

+) The proposed method is new and technically reasonale in general

+) Experimental results looks promising and justified the claims

**Weaknesses:**

-) Some technical details are not very clear

-) It is not clear if the conclusion obtained on classification task can be extended to a more general scope of models

**Questions:**

a) The focus on defending against false claim attacks is both timely and relevant, given the increasing reliance on open-source pre-trained models in diverse fields. By addressing the vulnerability in existing model fingerprinting methods, this paper provides an advancement in securing model ownership verification.

b) While I can understand the general idea of test sample extraction is to find a perturbation that pushes the model's prediction towards a target fingerprint (Eq.6), some technical details are not very clear to me. First, how to choose the target fingerprint F (Eq. 5 and 6)? I believe the choice of F significantly affects the model performance (e.g, F is different from the true label). Secondly, while the paper proposes two mapping functions f, it is not clear what criteria should be considered as good mapping functions. The design of FIT-MODELDIF and FIT-LIME looks arbitrary, and not clear why they are good mapping functions? Thirdly, I am confused how Eq.6 grantees that an independent model won't respond similarly to the test samples?

c) It is worth comparing to existing optimization-based model finger-printing methods, e.g, sensitive sample fingerprinting [1].

[1] He, Zecheng, Tianwei Zhang, and Ruby Lee. "Sensitive-sample fingerprinting of deep neural networks." Proceedings of the IEEE/CVF conference on computer vision and pattern recognition. 2019.

d) It would be better to discuss if the conclusion obtained on classification tasks can be extended to a more general scope of models, e.g., LLMs and other computer vision models (e.g., diffusion models which involves randomness)?

---

> ### Author Response · Authors · 2024-11-19
> **Author Response (Part I)**
>
> Dear Reviewer toL1, thank you very much for your careful review of our paper and thoughtful comments. We are encouraged by your positive comments on **the timely topic, novel and reasonable method, and promising experimental results** of our paper. We hope the following responses can help clarify potential misunderstandings and alleviate your concerns.
>
> ---
> **Q1**: The focus on defending against false claim attacks is both timely and relevant, given the increasing reliance on open-source pre-trained models in diverse fields. By addressing the vulnerability in existing model fingerprinting methods, this paper provides an advancement in securing model ownership verification.
>
>
> **R1**: Thank you for the positive comment! Your comment encourages us a lot.
>
>
> ---
> **Q2-1**: While I can understand the general idea of test sample extraction is to find a perturbation that pushes the model's prediction towards a target fingerprint (Eq.6), some technical details are not very clear to me. First, how to choose the target fingerprint F (Eq. 5 and 6)? I believe the choice of F significantly affects the model performance (e.g, F is different from the true label).
>
> **R2-1**: Thank you for the insightful comment! We are deeply sorry for the missing details in our submission that may lead to potential misunderstanding.
>
> - **In our method, the targeted fingerprint is a bit string representing the identity of the model developer and needs to be registered to the trustworthy verifier**. For instance, the logo of the company or personal identity number can be used as the targeted fingerprint.
> - **The choice of the targeted fingerprint does not affect the model performance**. This is because model fingerprinting does not alter the parameters of the models and has no impact on the model performance. This is a key advantage of fingerprinting.
> - In our main experiments, we utilize a logo of a file and a pen as the targeted fingerprint. We also conduct an ablation study utilizing three different fingerprints in *Appendix D.1*. The results in *Figure 7* demonstrate that **our FIT-Print is still effective with different targeted fingerprints**.
>
> We also provide more details in *Appendix F.1* of our revision.
>
> ---
>
> **Q2-2**: Secondly, while the paper proposes two mapping functions f, it is not clear what criteria should be considered as good mapping functions. The design of FIT-MODELDIF and FIT-LIME looks arbitrary, and not clear why they are good mapping functions?
>
> **R2-2**: Thank you for the insightful comment! We are deeply sorry that our submission may lead you to some misunderstandings that we want to clarify.
>
> - **FIT-ModelDiff and FIT-LIME are 'good' and outperform existing methods primarily due to the proposed FIT-Print paradigm.** Our FIT-Print leverages *targeted* fingerprinting to decrease the transferability of fingerprints and resist false claim attacks.
> - **FIT-ModelDiff and FIT-LIME are two concrete implementations under FIT-Print.**
>     - FIT-ModelDiff and FIT-LIME are **representatives of bit-wise method** (extract the fingerprint bit by bit) **and list-wise method** (extract the fingerprint as a whole), respectively.
>     - FIT-ModelDiff and FIT-LIME have different time and space complexity (details in *Appendix F*). For FIT-ModelDiff, the space complexity is $O(1)$ and the time complexity is $O(k)$ where $k$ is the length of the targeted fingerprint. For FIT-LIME, the space complexity is $O(k)$ and the time complexity is $O(k/\beta)$ where $\beta$ is the batch size.
> - **The design criteria for a 'good' mapping function are from four main aspects**:
>     - **Distinguishable**: Different models need to exhibit different outputs in the output space of the mapping function. This can guarantee that applying the mapping function can distinguish different independent models.
>     - **Task-agnostic**: The mapping function needs to be able to process the outputs of models with different tasks ($e.g.$, with different numbers of classes).
>     - **Robust**: The outputs of the mapping function on a model need to be robust against various model reusing techniques, $i.e.$, the outputs do not change significantly after model reusing.
>     - **Efficient**: The calculation of the mapping function needs to be efficient and take a small overhead.
>
> **We also provide more details on how to design new mapping functions under FIT-Print in *Appendix K***.

---

> ### Author Response · Authors · 2024-11-19
> **Author Response (Part II)**
>
> **Q2-3**: Thirdly, I am confused how Eq.6 guarantees that an independent model won't respond similarly to the test samples?
>
> **R2-3**: Thank you for the insightful comment! We will clarify the potential misunderstanding to alleviate your concern.
>
> - The core of making that an independent model won't respond similarly to the test samples is to **decrease the transferability of the fingerprint**.
> - **The key in our FIT-Print is the utilization of the targeted fingerprint and Eq. 6 aims to achieve such an objective.** Our main insight is that untargeted fingerprints enlarge the space of viable fingerprints while targeted fingerprints can decrease the transferability and reduce the probability of successful false claim attacks.
> - We admit that we do not have a strict theoretical guarantee that an independent model won't respond similarly. However, arguably, **such a phenomenon is also widely validated in the area of adversarial attacks ($i.e.$, targeted adversarial examples have lower transferability than untargeted ones).** The papers from the top peer-reviewed conferences [1,2] also present the proposition. It is also empirically validated as depicted in Table 1 of [3].
>
> **Reference**
> 1. Towards Transferable Targeted Attack. CVPR, 2020.
> 2. Towards Transferable Targeted Adversarial Examples. CVPR, 2023.
> 3. LFAA: Crafting Transferable Targeted Adversarial Examples with Low-Frequency Perturbations. arXiv, 2023.
>
>
> ---
> **Q3**: It is worth comparing to existing optimization-based model finger-printing methods, e.g, sensitive sample fingerprinting [1].
>
> [1] He, Zecheng, Tianwei Zhang, and Ruby Lee. "Sensitive-sample fingerprinting of deep neural networks." Proceedings of the IEEE/CVF conference on computer vision and pattern recognition. 2019.
>
> **R3**: Thank you for suggesting the outstanding related work! We will further clarify the similarities and differences between [1] and our work as follows.
>
> - **Similarities**
>     - [1] and our work both utilize model fingerprinting techniques which **do not need to fine-tune the model and have no impact on the model performance**.
>     - [1] and our work both depend on **optimizing the testing samples** to generate model fingerprints.
> - **Differences**
>     - **[1] focuses on a different task that utilizes fingerprinting for integrity verification of models.**
>         - [1] attempts to verify whether the model parameters are altered by the adversary while our work focuses on verifying whether a suspicious model is a reused version of another model.
>         - [1] leverages fragile fingerprinting while our method is a robust fingerprinting method.
>     - **The technique proposed in [1] cannot be directly adapted to our scenario.** [1] aims to generate a fragile fingerprint that can be destroyed when the model is modified by others. However, in our scenario, directly applying [1] may lead to that the fine-tuned, pruned, and extracted models will not be regarded as reused, since the (fragile) fingerprints inside these models are destroyed and no longer similar to the fingerprint of the original model. It will inevitably lead to a high false negative rate.
>
> Despite the differences, [1] is still an outstanding work exploring another application (integrity verification) of model fingerprinting. We will add a discussion of [1] in *Appendix L.2*. If you have suggestions on other optimization-based fingerprinting methods, please let us know. We are also willing to provide a discussion of them :)

---

> ### Author Response · Authors · 2024-11-19
> **Author Response (Part III)**
>
> **Q4**: It would be better to discuss if the conclusion obtained on classification tasks can be extended to a more general scope of models, e.g., LLMs and other computer vision models (e.g., diffusion models which involves randomness)?
>
> **R4**: Thank you for the insightful comment! We admit that the generalization of our FIT-Print to a more general scope of models is important. We will further clarify it as follows.
>
> - Arguably, for deterministic models which do not involve randomness ($e.g.$, CNN and LLM), **our FIT-Print can generalize to different types of models and datasets** (details in *Appendix H*).
>     - **Our FIT-Print can generalize to models with different architectures and tasks**.
>         - Since we do not need to alter or fine-tune the model, **our method can fundamentally generalize to models** with other architectures ($e.g.$, advanced CV models like ViT) as well.
>         - For models with different tasks, **the major difference between models with different tasks is the output format**. For instance, the image generation model outputs a tensor consisting of a sequence of logits. FIT-ModelDiff calculates the cosine similarity between the outputs and FIT-LIME calculates the average entropy of the output. The two calculation methods can be applied to any output format ($e.g.$, 1-D vectors, 2-D matrices, or tensors). As such, **our methods are naturally feasible for models with different tasks**.
>     - **Our FIT-Print can generalize to different types of datasets**.
>         - For different datasets (particularly discrete data like text data), the main challenge of optimizing the testing samples lies in how to design an effective optimization method. There are already some existing works[1,2,3] to fulfill this task. Accordingly, **our FIT-Print can be adapted to other data formats ($e.g.$, text or tabular)**.
>         - **We also conduct a case study on text generation models in *Appendix H.3***. The experimental results demonstrate that our FIT-Print still works for text generation models.
> - For non-deterministic models that involve randomness ($e.g.$, diffusion models), we have to admit that we do not know whether our methods and existing fingerprinting methods can be adapted to them since they have a completely different inference paradigm. We will conduct a comprehensive study in our future work.
>
> **Reference**
>
> 1. Gradient-based adversarial attacks against text transformers. EMNLP, 2021.
> 2. Hard prompts made easy: Gradient-based discrete optimization for prompt tuning and discovery. NeurIPS, 2023.
> 3. Promptcare: Prompt copyright protection by watermark injection and verification. S&P, 2024.

---

> ### Author Response · Authors · 2024-11-23
> **Thanks to Reviewer toL1**
>
> Please allow us to thank you again for reviewing our paper and the valuable feedback, and in particular for recognizing the strengths of our paper in terms of the timely topic, novel and reasonable method, and promising experimental results.
>
> Please let us know if our response has properly addressed your concerns. We are more than happy to answer any additional questions during the discussion period. Your feedback will be greatly appreciated.

---

> > ### Author Response · Authors · 2024-11-25
> > **A Gentle Reminder of the Post-rebuttal Feedback**
> >
> > Thank you very much again for your initial comments. They are very valuable for improving our work. We would be grateful if you could have a look at our response and modifications and please let us know if there is anything else that can be added to our next version. We are also willing to have further discussions with you.

---

> > > ### Author Response · Authors · 2024-11-28
> > > **A Second Reminder of the Post-rebuttal Feedback**
> > >
> > > Dear Reviewer toL1,
> > >
> > > We greatly appreciate your initial comments. We totally understand that you may be extremely busy at this time. But we still hope that you could have a quick look at our responses to your concerns. We appreciate any feedback you could give us. We also hope that you could kindly update the rating if your questions have been addressed. We are also happy to answer any additional questions before the rebuttal ends.
> > >
> > > Best Regards,
> > >
> > > Paper 5867 Author(s)

---

> ### Author Response · Authors · 2024-12-02
> **Reminder of the Post-rebuttal Feedback and Summary of Our Response**
>
> Dear Reviewer toL1,
>
> Thank you for your time and effort in evaluating our work. We greatly appreciate your initial comments. Your insights and suggestions are extremely valuable to us.
>
> Given that we have only *one day* left for discussion, we are hoping to receive any additional feedback or question you might have at your earliest convenience. We totally understand that you may be busy at this time. But we still hope that you could have a quick look at our responses to your concerns. Your expertise would be of great help to us in improving the quality and rigor of our work.
>
> To facilitate the discussion, we would like to summarize our response as follows.
>
> - **We clarified and added more technical details of our method**, including how to choose the target fingerprint $F$, how to design a good mapping function, and how and why the loss function (Eq. 6 in our paper) works.
> - **We discussed the suggested related work [1]**. [1] focuses on a different task that utilizing fingerprinting for model integrity verification.
> - **We also discussed the generalization of our method to a more general scope of models**, such as LLMs and diffusion models.
>
> If our responses address your concerns, we kindly request that you reconsider your evaluations. We would also be grateful for any additional comments or suggestions you might have to refine our work.
>
> Best regards,
>
> Paper 5867 Author(s)
>
> **Reference**
>
> 1. Sensitive-sample fingerprinting of deep neural networks. CVPR, 2019.

---

### Author Response · Authors · 2024-11-19
**Concerns Regarding the Potential LLM-written Review from Reviewer LMaC**

Dear Chairs and Reviewers:

Greetings and wish you all the best. We deeply and sincerely thank you and all reviewers for your valuable time and comments on our paper. Your efforts greatly help us to improve our paper.

In this letter, we would like to express our concern regarding **the potential completely LLM-written review of Reviewer LMaC**. We exploit three different AI content detectors ([[link1]](https://quillbot.com/ai-content-detector), [[link2]](https://copyleaks.com/ai-content-detector), [[link3]](https://sapling.ai/ai-content-detector)) from the Internet to check whether the review is AI-generated. The results show that **over 94% of contents in the review of Reviewer LMaC is AI-generated**.

More importantly, **this review is highly biased and even erroneous**. Some examples are as follows:

- The review of Reviewer LMaC **simply negates our contributions** by omitting our discussions, explanations for our insight in Section 1, and our experiments which demonstrate our contributions in Section 4.
- **Some comments of the review are unreasonable and unfounded**.
    - Reviewer LMaC comments that naming a method can only use the first letter and requires us to explain some common-sense mathematical terms such as 'Theorem' and 'Definition'.
    - Reviewer LMaC claims that we overuse the abbreviation 'p1 and p2'. However, we actually use the full name 'Proposition 1 and Proposition 2' in our paper.

**We sincerely hope that you can take notice of these issues and ignore his/her follow-up comments to our paper**. We believe that this reviewer might undermine the impartiality and fairness of the ICLR reviewing process, although we believe that we have addressed his/her concerns in our rebuttal. More detailed justifications are as follows:

- Considering that this reviewer gave us an extremely negative score of 1 and that his review comments were entirely generated by LLM, **we have to worry about whether he/she had a preconceived extremely negative stance on our work from the very beginning**, and even purposely and directly asked LLM to write a very negative comment for the purpose of rejecting the paper in bad faith.
- As emphasized in the email to the ICLR reviewers, '**the use of LLMs to write full reviews is not allowed, LLM written reviews are easy to spot and the program committee will be rejecting such reviews**'.
- Considering his/her previous negative and evil preconceived position and our accusations against him/her, **he/she might deliberately conduct malicious guidance in the subsequent discussion to cover up his/her malicious purpose and complete his/her malicious rejection of our manuscript**.

**We respect the opinions of the reviewers, but we sincerely hope that our work will be treated objectively and fairly**. We have poured our hearts and souls into this work, and we believe it deserves to be treated fairly. We are deeply sorry for the inconvenience that our notifications may cause you. Thank you again for all your kind and valuable efforts in the whole reviewing process.

Best Regards,

Paper 5867 Author(s)

---

### Comment · Area_Chair_qhR4 · 2024-11-23

Dear Reviewers,
The authors have responded to your valuable comments.
Please take a look at them!

Best,
AC

---

### Author Response · Authors · 2024-11-27
**Thanks to Reviewer and A Gentle Reminder of Discussion**

Dear Reviewers,

We thank Reviewer *jEMa* for the discussion and raising the score. As we have less than two days to further revise our paper, we would like to kindly remind Reviewers *toL1* and *oTwh* to take a look at our responses. We sincerely appreciate your time and any feedback you could give us.

---

### Meta-Review · Area_Chair_qhR4 · 2024-12-13

**Metareview:**

In the work, the authors studied ``Model fingerprinting,'' which was adopted to verify whether a suspicious model is reused from the original source one.

The authors have the concern regarding the comments from Reviewer LMaC that may be completely LLM-written (although the AC has no 100% evidence).

Therefore, the decision is made without considering Reviewer LMaC's comments.

During the rebuttal process, the authors have addressed more comments from the reviewers except that a theoretical analysis is still a lack.

In particular, although the Reviewer jEMa raised the score,  this work still cannot provide theoretical support.
Specifically, Reviewer jEMa concerns ``the current manuscript lacks a deeper analysis or evaluation on the transferability of the targeted fingerprints. ... However, it is unclear whether a more sophisticated adversary could forge a false claim on FIT-Print (e.g., by exploiting techniques from transferrable targeted adversarial attacks). Apart from a lack of formal proof (which is a noteworthy but understandable limitation, given the complexity of neural networks), the current adaptive evaluation is also limited (in Section 4.4, the adaptive attacker only leverages additional independent models). Hence, a more in-depth analysis or evaluation on this aspect would help make the claim more persuasive.''
The AC agrees with the point and this submission at its current status did not meet the high standard of ICLR.

**Additional Comments On Reviewer Discussion:**

The authors still cannot provide theoretical support for their work.

The AC has also contacted with Reviewer LMaC, who was assigned to review this paper by the Review System instead of the AC.
Reviewer LMaC provided concrete evidences that the review comments came from him/her but were polished by ChapGPT.

---

### Decision · Program_Chairs · 2025-01-22

Reject